

# Stress induced on permanent mandible first molar and space maintainer under normal masticatory forces: a finite element study

Hui Shi[1], Fang Fang Kang[2] and Qian Liu[3]

[1] Department of Orthodontics, Nantong Stomatological Hospital Affiliated to Nantong University, Nantong, Jiangsu, China
[2] Department of Stomatology, The Second Affiliated Hospital of Nanjing Medical University, Nanjing, Jiangsu, China
[3] Department of Pediatric and Preventive Dentistry, The Affiliated Stomatological Hospital of Nanjing Medical University, Nanjing, Jiangsu, China

Corresponding authors
Fang Fang Kang, kff@njmu.edu.cn
Qian Liu, 8817070@qq.com

## ABSTRACT

**Background:** The band and loop space maintainer is used to maintain the missing space of deciduous molars which are lost early. When the second deciduous molar is lost prematurely, the stress on the first permanent molar during different degrees of development may vary when it is the abutment. The design and use of the space maintainer may also lead to damage of the loop. The purpose of this article is to use the finite element method to study the stress on the first permanent molar and the loop with or without occlusal contact, with the first permanent molar of four different degrees of development serving as the abutment. We aimed to guide the clinical design and use of the space maintainer.

**Methods:** We developed finite element models of the mandibular first permanent molar and the band and loop space maintainer, and simulated alveolar bone, periodontal ligament (PDL), enamel and dentin. The four developmental stages were 1/2 (I), 2/3 (II), 3/4 (III) and full development (IV). Ansys Workbench was used to analyze the effects of root development and occlusal contact between the loop and the opposite jaw on abutment teeth and the loop. Abutment teeth were statically loaded vertically and obliquely with a force of 70 N. The loop was statically loaded vertically with a force of 14 N. The stress on all structures and the displacement trends of the loop were calculated.

**Results:** The stress on enamel, dentin, PDL and alveolar bone were similar, and the concentration was consistent. But if there was occlusal contact, the loop produced maximum displacement at the near middle edge of contact with the anterior teeth. When the loop was in occlusal contact with the opposing occlusal tooth, the peak value of the equivalent stress on the space maintainer under vertical load was: group I > group IV > group III > group II, and the maximum principal stress peak change was: group I > group III > group II > group IV. The change of the equivalent stress peak value of the loop under oblique load was: group I > group III > group IV > group II, and the maximum principal stress peak change was: group III > group I > group II > group IV. When the loop was not in occlusal contact with the opposing occlusal tooth, the peak value of the equivalent stress on the space maintainer under vertical load was: group IV > group I > group II > group III, and the maximum principal

stress peak change was: group IV > group I > group II > group III. The change of the equivalent stress peak value of the space maintainer under oblique load was: group I > group IV > group II > group III, and the maximum principal stress peak change was: group I > group IV > group II > group III.

**Conclusions:** Our results suggested that whenever possible, choosing the teeth with nearly complete root development as the abutment of the space maintainer is advisable. The design and use of the band and loop space maintainer should avoid occlusal contact with the occlusal teeth to prevent deformation of the loop.

## INTRODUCTION

Severe deciduous dental caries, periapical disease, and trauma can lead to premature loss of deciduous teeth, resulting in the irregular arrangement of permanent teeth due to insufficient eruption space for permanent teeth. Similarly, the early loss of young permanent teeth also causes displacement of adjacent teeth, leading to malocclusion (*Wojtaszek-Lis et al., 2018*; *American Academy of Pediatric Dentistry, 2017*). Therefore, after the early loss of teeth in children, a space maintainer is typically used to maintain the mesial and distal distance of the missing teeth in order to promote the normal eruption of permanent teeth. This prevents the adjacent teeth from tilting towards the gap and the opposite jaw teeth from extending. The band and loop space maintainer is suitable for the premature loss of unilateral or bilateral single deciduous molars, or the premature loss of the second deciduous molars and the complete eruption of the first permanent molars (*Pawar, 2019*). When the second deciduous molar is lost early, it is necessary to use the space maintainer to preserve the missing space in order to maintain a better arch length. Simultaneously, the mesial displacement of the first permanent molar is more obvious, so the clinical use of the band and loop space maintainer is necessary.

In the mixed dentition stage, the first permanent molar is still developing, meaning its root is not fully developed. When the band and loop space maintainer is applied to the first permanent molar, its root may not be mature, and the stress it bears may have an adverse effect on root development. Studying the relationship between the band and loop space maintainer and stress on immature teeth is helpful for improving the selection and design of the space maintainer.

Finite element analysis was first proposed by *Turner (1956)*. This method divides a continuous entity into finite elements and replaces the original continuum with the combination of each element. The mechanical properties of each element are studied one by one and the stiffness balance equation of each element is established. Then the group of the overall stiffness equilibrium equation is integrated, and the overall stiffness balance equation is solved according to the given boundary displacement condition and load condition. The displacements of all the nodes of the element are obtained, and the internal forces and stress on the element are calculated accordingly, resulting in the mechanical

properties of the whole composite. Currently, finite element analysis has become a numerical "virtual experiment," replacing many physical experiments. The combination of many computational analyses based on this method and typical confirmatory experiments can achieve high efficiency at a low cost (*Se-Young et al., 2017*; *Zheng et al., 2016*).

The purpose of this article is to use the finite element method to analyze the stress on the first permanent molar and the loop when the band and loop space maintainer is applied to the first permanent molars at different developmental stages.

## MATERIALS AND METHODS

### Equipment

Dell Precision T5820 Desktop: CPU E5-2643; win10 64-bit; 16G RAM; 1T HDD; GraphicsNVIDIA Quadro 4000
Mimics 19.0 (Materialize Inc. Belgium)
Proe5.0 (Parametric Technology Corporation, Boston, Massachusetts, USA)
Geomagic Studio 2013 (Geomagic Studio Product, Morrisville, North Carolina)
UGNX8.5 (Siemens PLM Software, Plano, Texas, USA)
Ansys Workbench (ANSYS Inc, Canonsburg, Pennsylvania, USA)

### Methods

A 7-year-old child with mixed dentition was selected for this study, and the mandible of the subject was scanned by cone beam CT (CBCT data from this 7-year-old child who needed other treatments, and the mandibular first permanent molar has no caries and no fillings). The study protocol was approved by the Ethics Committee of Nantong Stomatological Hospital (approval number: 2022-003-01). Written informed consent was obtained from the parents or guardians of children. The image acquisition parameters were: tube voltage 10 kV; tube current 20 mA; scan time 26 s; layer thickness 0.2 mm. The field of view and three-dimensional (3D) body velocities were $15 \times 15$ cm and $0.3 \times 0.3$ and 0.3 mm, respectively. A total of 377 tomographic images were obtained, stored, and engraved in standard DICOM format. The CT image was imported into Mimics19.0 in DICOM format, and the three-dimensional model of the mandibular alveolar bone was established by Mimics (Fig. 1). The model was inputted into Geomagic Studio in *.stl format, and the protuberances and miscellaneous spots on the model surface were removed in Geomagic Studio to make the model surface smoother. First, a three-dimensional model of normal mandibular first permanent molars with fully developed roots was created, followed by models of mandibular first permanent molars in the 1/2, 2/3, and 3/4 stages of root development. Then, the root of the model was separated from the crown along the alveolar bone and along its crown. The periodontal ligament was formed by expanding 0.15 mm using the thickening command, and a band and loop space maintainer with the abutment of the mandibular first permanent molar was designed. Finally, using the curved suture function, the first permanent molars, alveolar bone, periodontal ligament, and band and loop space maintainer were converted into UGNX in *.igs format to form a solid model
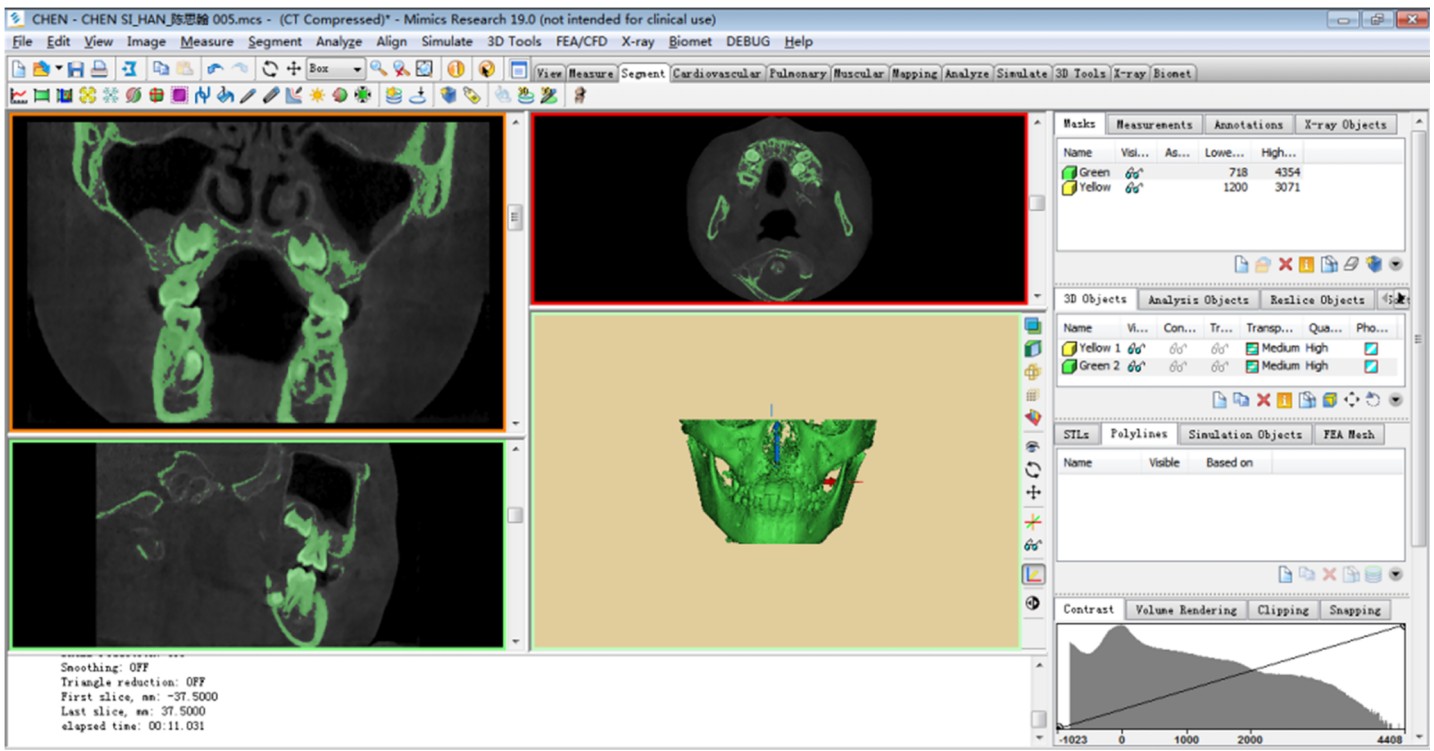

**Figure 1 Preliminary 3D model.** A total of 377 tomographic images were obtained, stored, and engraved in standard DICOM format. The CT image was imported into Mimics19.0 in DICOM format, and the three-dimensional model of the mandibular alveolar bone was established by Mimics.

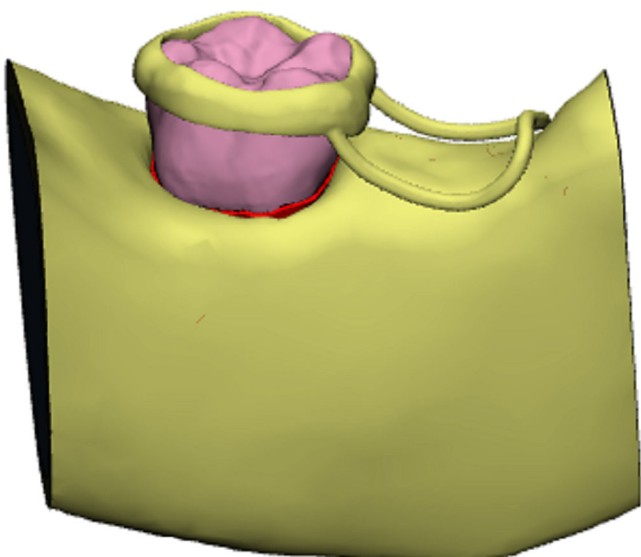

**Figure 2 Three-dimensional model of mandibular first permanent molar and the band and loop space maintainer.**

(Fig. 2). The solid models of the four types of first permanent molars and band and loop space maintainer required for the experiment were built.

Four types of 3D solid models created in UGNX were imported into Proe in the *.stp format. In Proe, the band and loop space maintainer was attached to the crown of the first permanent molar, completing the structure. The assembled model was saved as an *.asm file, enabling us to construct a three-dimensional solid model. This model depicted the first permanent molar with four different degrees of root development as abutment teeth for the band and loop space maintainer (Fig. 3).

The constructed 3D solid models were inputted into the Workbench analysis interface through the connection between Proe and Ansys Workbench finite element analysis software. The element division of the model was carried out using the automatic meshing function of Workbench, resulting in the three-dimensional finite element model.

## Finite element stress analysis

### Definition of material and boundary conditions

It was assumed that the relative displacement of the interface between tooth and periodontal ligament, periodontal ligament and alveolar bone, and band and loop space maintainer in the model was zero, which was defined as a full constraint relationship. The interface between the band and loop space maintainer and the first permanent molar was defined as bonding. We applied a y-direction constraint to the loop, and the alveolar bone base was set as a fixed constraint. The materials and structures in the model were assumed to be isotropic homogeneous linear elastic materials, and the mechanical properties of each material are shown in Table 1.

### Load conditions

*First permanent molar*

The static loading force was set to be uniformly distributed, and the loading directions were vertical and oblique.

Vertical loading: the direction was 15° to the long axis of the tooth, and the loading sites were the mesial marginal crest, central fossa, distal marginal crest, mesial buccal tip, and distal buccal tip. The total loading force was 70 N, and the force value of each loading site was 14 N.

Oblique loading: the direction was 60° to the long axis of the tooth, and the loading point was the middle point of the oblique surface of the proximal and distal buccal tips. The total loading force was 70 N, and the force value of each loading site was 35 N.

*Loop*

To apply a static load, we uniformly distributed a 14 N force vertically at the contact edge between the loop and the opposite jaw. The loading sites and loading directions discussed above are shown in Fig. 4.

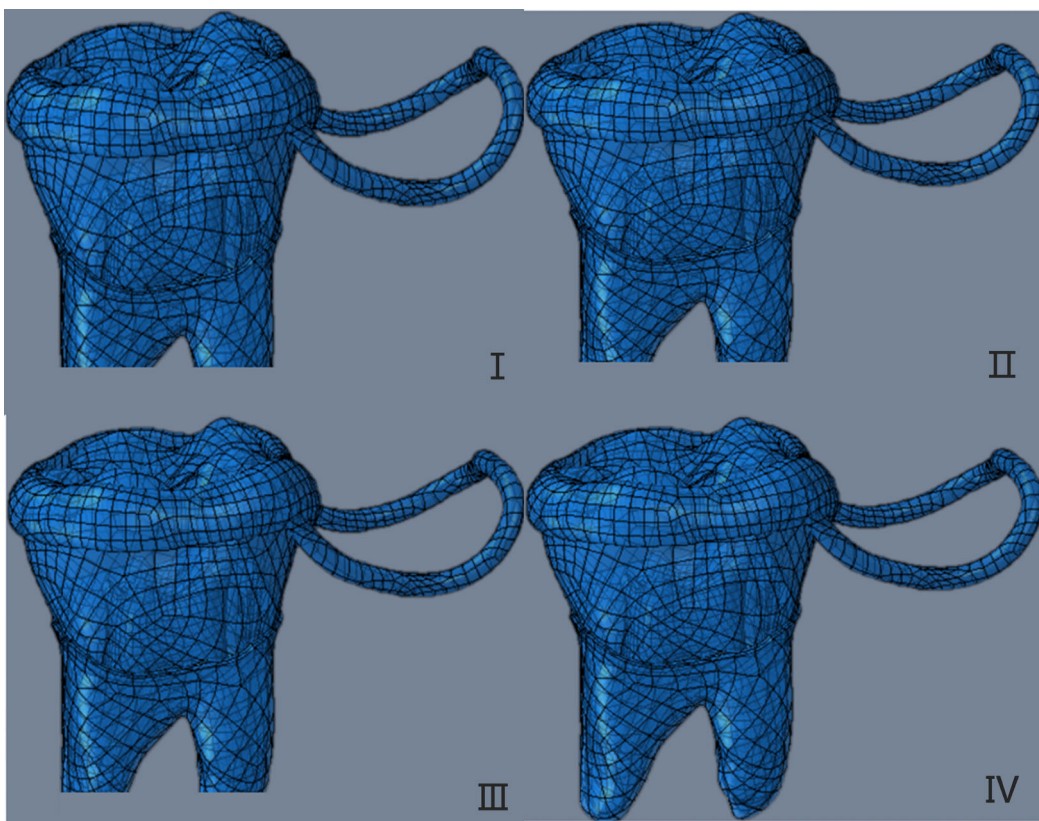

**Figure 3 The three-dimensional solid model of the first permanent molar with four different root development degrees as abutment teeth as the band and loop space maintainer.** The four developmental stages were 1/2(I), 2/3(II), 3/4(III) and fully developed (IV).

**Table 1 Mechanics parameter of materials.**

|  | Modulus of elasticity (MPa) | Poisson's ratio |
|---|---|---|
| Alveolar cortial bone | 13,700 | 0.3 |
| Alveolar cancellous bone | 1,370 | 0.3 |
| PDL | 69 | 0.45 |
| Dentine | 18,600 | 0.31 |
| Enamel | 84,000 | 0.33 |
| Pulp | 2 | 0.45 |
| Stainless steel | 194,020 | 0.3 |

## RESULTS

In the two cases of occlusal contact between the loop and the opposite jaw, the maximum equivalent stress and the first principal stress on the four models of root development are shown in Table 2. The stress concentration areas of enamel and PDL are shown in Table 3. The equivalent stress, maximum principal stress distribution, and displacement clouds of enamel, dentin, PDL and the space maintainer are shown in Figs. 5–22. In the stress and

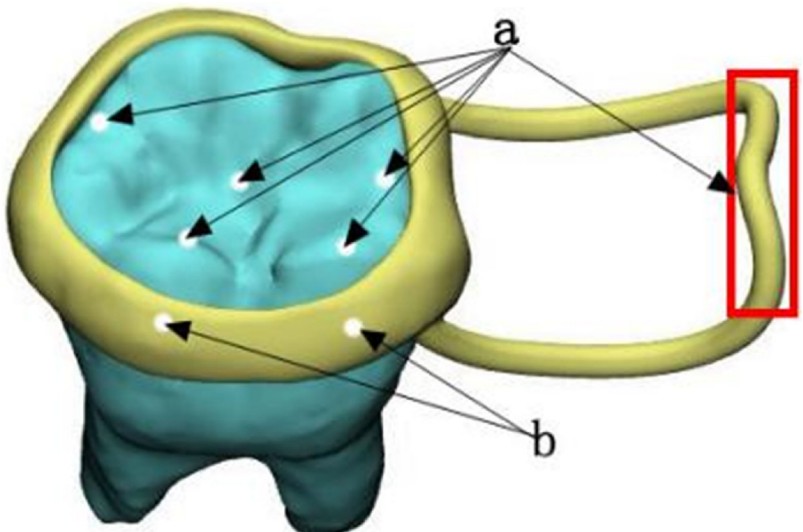

**Figure 4 Loading sites and loading directions.** (A) Vertical decentralized loading location; (B) obliquely dispersed loading position.

displacement cloud diagram, the size of the value is represented by color. Red, yellow, green, and blue are used in descending order.

The main results were as follows:

(1) The stress on enamel, dentin, periodontal ligament and alveolar bone were similar, and the stress concentration was consistent.

(2) According to the degree of root development of the four kinds of teeth, the maximum equivalent stress and the first principal stress on enamel and the periodontal ligament were the highest in group IV, while the stress on dentin was the lowest in group IV. The maximum equivalent stress and principal stress on dentin in group IV were less than in the other three groups.

(3) There were no significant differences in the equivalent stress and the first principal stress on enamel among the three groups when the root development was 1/2, 2/3 and 3/4.

(4) When the loop was in occlusal contact with the opposing occlusal tooth, the peak value of the equivalent stress on the space maintainer under vertical load was: group I > group IV > group III > group II, and the maximum principal stress peak change was: group I > group III > group II > group IV. The change of the equivalent stress peak value of the loop under oblique load was: group I > group III > group IV > group II, and the maximum principal stress peak change was: group III > group I > group II > group IV. When the loop was not in occlusal contact with the opposing occlusal tooth, the peak value of the equivalent stress on the space maintainer under vertical load was: group IV > group I > group II > group III, and the maximum principal stress peak change was: group IV > group I > group II > group III. The change of the equivalent stress peak value of the space maintainer under oblique load was: group I > group IV >

**Table 2 With or without occlusal contact between the loop and the opposite jaw, the maximum equivalent stress and the first principal stress (MPa) of each model.**

| Model | | Without contact | | | | With contact | | | |
|---|---|---|---|---|---|---|---|---|---|
| | | Maximum equivalent stress | | Maximum first principal stress | | Maximum equivalent stress | | Maximum first principal stress | |
| Root development | | Vertical | Oblique | Vertical | Oblique | Vertical | Oblique | Vertical | Oblique |
| Space maintainer | 1/2 | 74.8 | 268.2 | 60.9 | 201.9 | 420.6 | 397.0 | 431.7 | 368.0 |
| | 2/3 | 53.3 | 148.3 | 44.8 | 110.7 | 382.0 | 367.5 | 382.9 | 343.5 |
| | 3/4 | 49.3 | 121.9 | 44.0 | 105.1 | 382.3 | 374.5 | 428.0 | 394.4 |
| | 1 | 106.9 | 261.6 | 80.8 | 194.6 | 388.6 | 368.3 | 376.7 | 328.5 |
| Enamel | 1/2 | 44.4 | 113.5 | 18.0 | 110.6 | 44.4 | 109.4 | 18.1 | 110.7 |
| | 2/3 | 44.4 | 102.5 | 12.2 | 110.5 | 44.4 | 102.3 | 12.1 | 110.7 |
| | 3/4 | 44.4 | 101.3 | 12.2 | 110.0 | 44.4 | 101.0 | 12.1 | 110.2 |
| | 1 | 82.2 | 193.4 | 28.5 | 120.2 | 90.1 | 196.9 | 32.0 | 110.8 |
| Dentine | 1/2 | 34.9 | 172.9 | 21.2 | 108.9 | 38.8 | 165.0 | 23.0 | 100.2 |
| | 2/3 | 16.8 | 69.7 | 15.1 | 65.7 | 16.6 | 64.4 | 14.4 | 60.6 |
| | 3/4 | 24.7 | 118.5 | 14.2 | 81.2 | 30.9 | 107.3 | 20.7 | 77.1 |
| | 1 | 5.4 | 14.9 | 4.8 | 16.5 | 6.3 | 13.8 | 4.0 | 15.8 |
| PDL | 1/2 | 5.2 | 13.0 | 2.1 | 9.5 | 4.4 | 12.1 | 2.2 | 9.0 |
| | 2/3 | 2.0 | 6.7 | 1.1 | 4.3 | 1.5 | 5.5 | 1.1 | 3.8 |
| | 3/4 | 1.0 | 2.4 | 0.7 | 3.1 | 1.2 | 2.0 | 0.7 | 2.8 |
| | 1 | 18.3 | 24.5 | 6.6 | 28.8 | 19.2 | 27.4 | 6.3 | 26.0 |
| Alveolar bone | 1/2 | 3.0 | 14.7 | 2.7 | 11.2 | 2.8 | 14.9 | 2.4 | 11.1 |
| | 2/3 | 2.0 | 4.4 | 2.5 | 5.1 | 1.9 | 4.4 | 2.3 | 4.7 |
| | 3/4 | 3.1 | 4.5 | 3.5 | 4.2 | 3.0 | 4.2 | 3.5 | 3.9 |
| | 1 | 4.9 | 6.0 | 4.1 | 5.2 | 4.4 | 5.5 | 3.5 | 4.7 |

group II > group III, and the maximum principal stress peak change was: group I > group IV > group II > group III.

(5) When observing the overall force of the band and loop space maintainer, the stress was concentrated on the edge of the contact between the loop and the band when there was no occlusal contact between the loop and the opposite jaw teeth. When the loop had poor contact with the opposite jaw, the stress was concentrated in the bend in the middle of the loop (Fig. 22).

(6) There was occlusal contact between the loop and the jaw teeth, and the maximum displacement appeared at the edge of the contact between the loop and the first deciduous molars.

## DISCUSSION

Space maintainers are widely used after the premature loss of deciduous teeth. Of the various types of fixed space maintainers, band and loop space maintainers are the most

**Table 3 The stress concentration area for each model.**

| Model | Load direction | With or without occlusal contact | Stress concentration area of equivalent stress | | Stress concentration area of first principal stress | |
|---|---|---|---|---|---|---|
| | | | Enamel | PDL | Enamel | PDL |
| I | Vertical | Without | Occlusal force loading zone | Mesial neck, buccal mid-neck | Occlusal force loading zone | Buccal side of the buccal root |
| | Vertical | With | | Apex, buccal side neck | | Buccal side of the buccal root |
| | Oblique | Without | | Buccal side of the lingual root, buccal side of the buccal root | | Buccal side of the distal lingual root |
| | Oblique | With | | Buccal side of the lingual root, buccal side of the buccal root | | Buccal side of the distal lingual root |
| II | Vertical | Without | | Buccal side of the lingual root and buccal root | | Mesial buccal side neck |
| | Vertical | With | | Mesial neck, upper root | | Mesial buccal side neck |
| | Oblique | Without | | Buccal side neck, buccal side of the mesial buccal root and distal lingual root | | Distal lingual side neck |
| | Oblique | With | | Buccal side neck, buccal side of the mesial buccal root and distal lingual root | | Distal lingual side neck |
| III | Vertical | Without | | Mesial neck, upper buccal root | | Upper root |
| | Vertical | With | | Buccal side neck, upper root | | Upper root |
| | Oblique | Without | | Buccal side neck | | Distal lingual side neck, lingual side of mesial buccal root |
| | Oblique | With | | Buccal side neck | | Distal lingual side neck, lingual side of mesial buccal root |
| IV | Vertical | Without | | Neck, apex | | Distal lingual side neck |
| | Vertical | With | | Neck, apex | | Neck of distal lingual root |
| | Oblique | Without | | Buccal side of the mesial buccal root and distal lingual root | | Neck of distal lingual root |
| | Oblique | With | | Buccal side of the mesial buccal root and distal lingual root | | Neck of distal lingual root |

frequently used (*Tuka & Heidrun, 2023*). This article focuses on the band and loop space maintainer, which is suitable for unilateral or bilateral premature loss of single deciduous molars, or premature loss of second deciduous molars with complete eruption of the first permanent molars. In mixed dentition stage, severe periapical lesions of the second

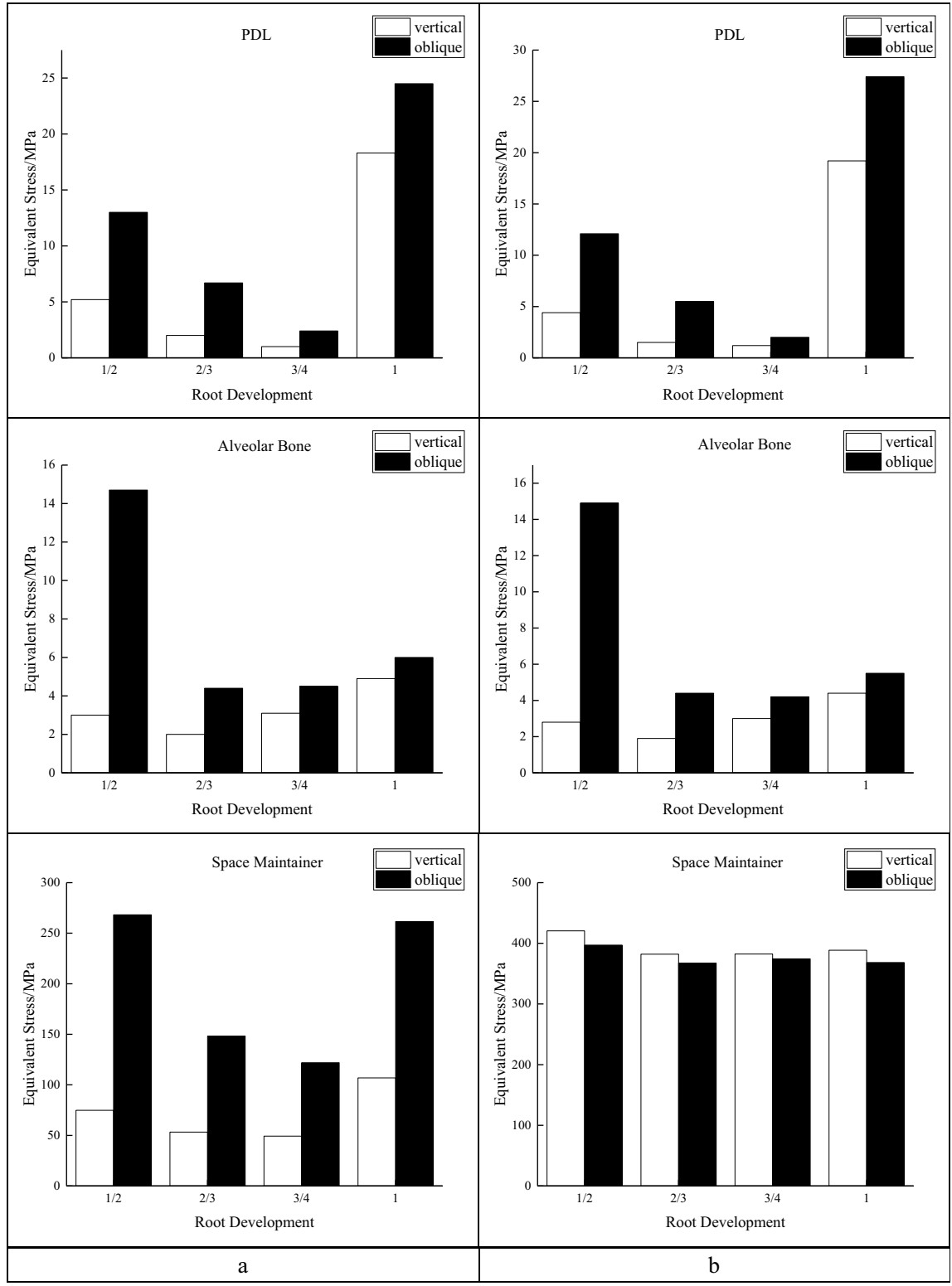

**Figure 5** (Continued)

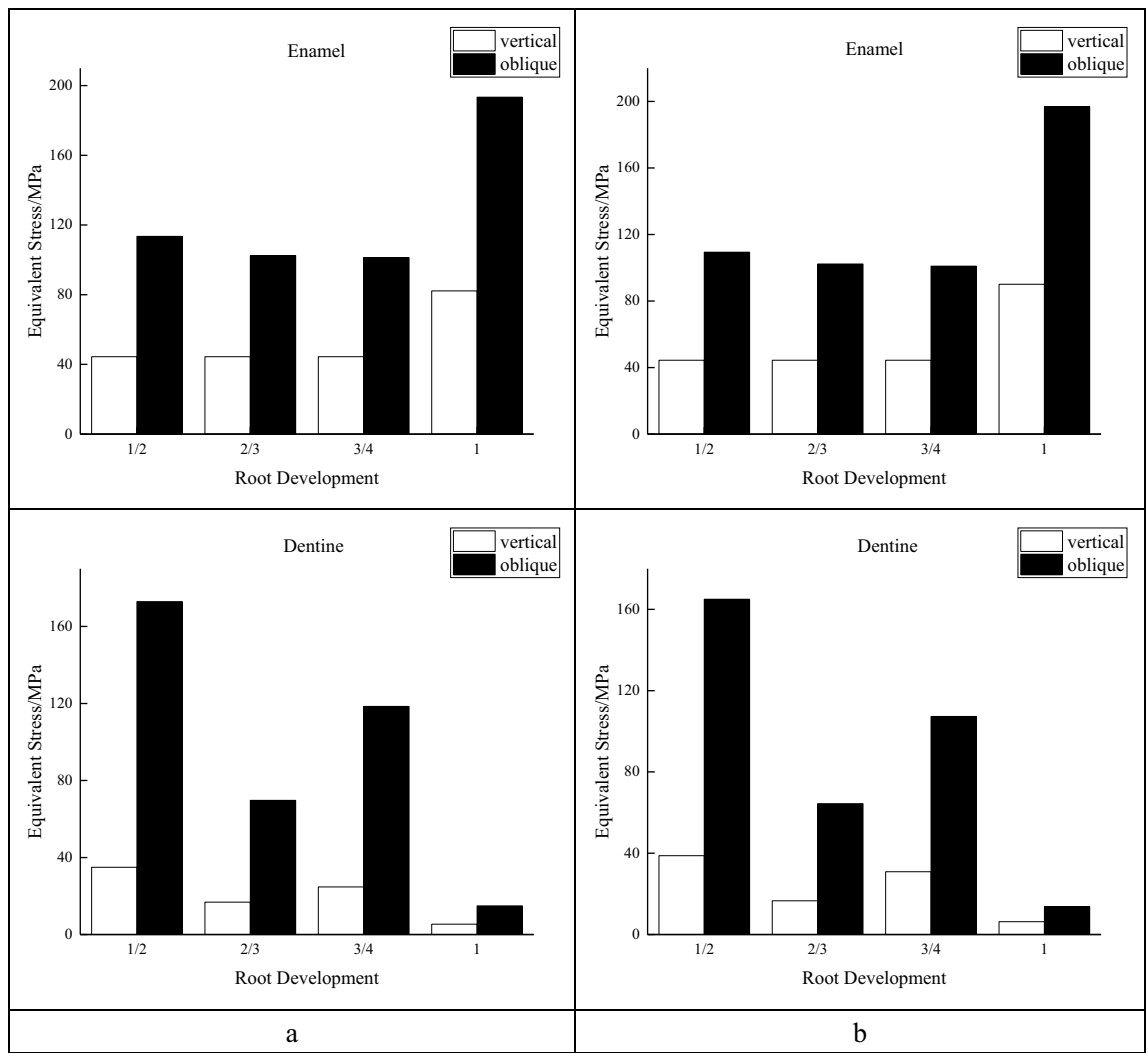

**Figure 5 The equivalent stress.** (A) Without contact; (B) with contact.

deciduous molars can lead to the early loss of the second deciduous molars and the early eruption of the first permanent molars (*Almeedani et al., 2020*; *Reddy et al., 2018*; *Al-Shahrani et al., 2015*; *Heilborn et al., 2011*). Because the first permanent molars erupt early, their root development is not yet complete, and there may be four stages of root development after the eruption. In this article, the first permanent molars in four stages of development were used as the abutment teeth to study the stress on the first permanent molars and the space maintainer with and without poor occlusal contact between the loop and the opposite jaw teeth.

This study is an exploratory study in the field of space maintainers. It examines the possible root development of four types of teeth after the eruption of the first permanent molar and analyzes the force on the teeth and the space maintainer when the first permanent molars are used as the base teeth to make a band and loop space maintainer. It also investigates scenarios where the loop does or does not produce poor bite contact

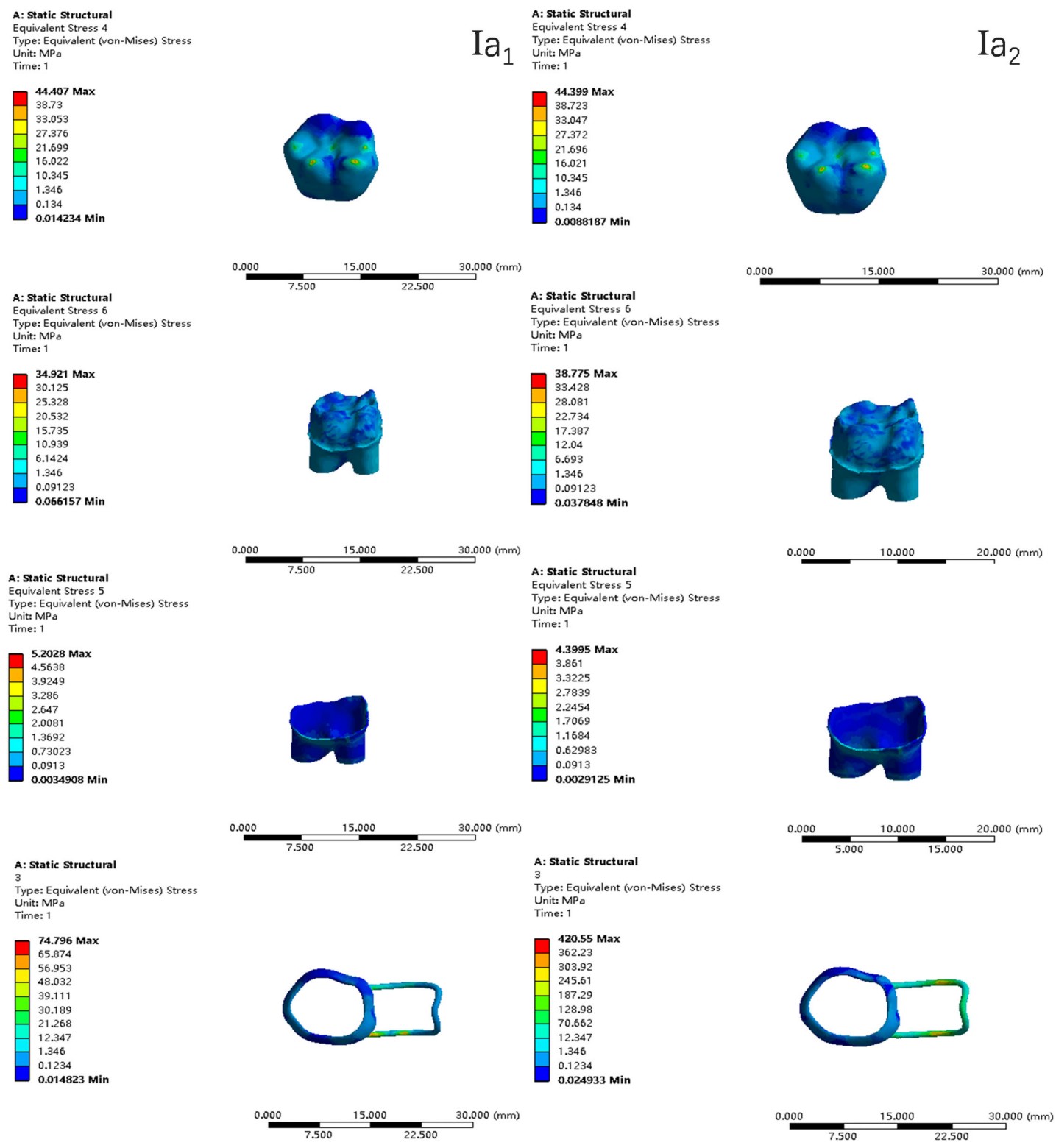

**Figure 6 The equivalent stress distribution clouds (I: root development of 1/2; a: vertical loading; 1: without occlusal contact; 2: with occlusal contact).**

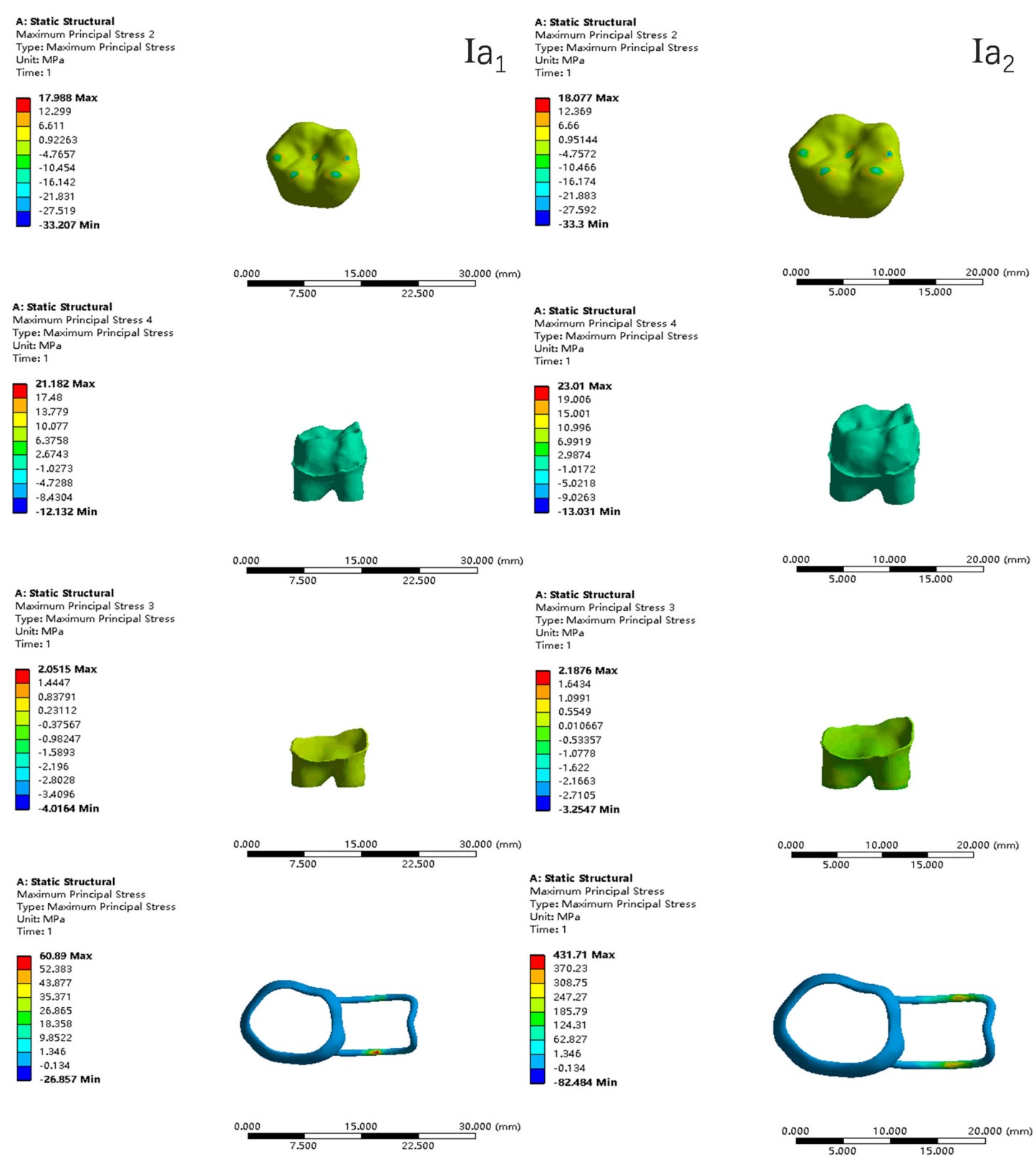

**Figure 7** The maximum principal stress distribution clouds (I: root development of 1/2; a: vertical loading; 1: without occlusal contact; 2: with occlusal contact).

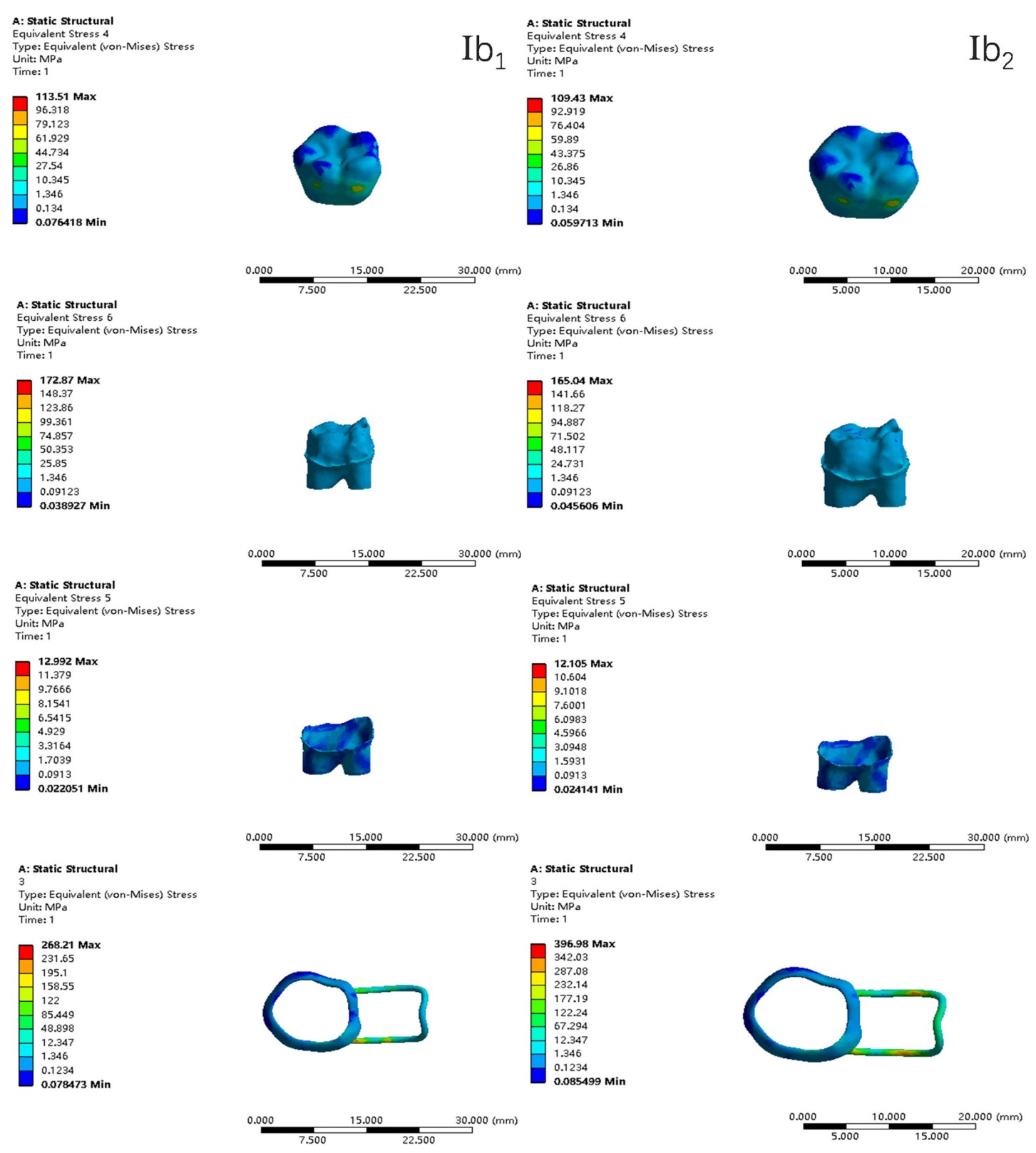

**Figure 8** The equivalent stress distribution clouds (I: root development of 1/2; b: oblique loading; 1: without occlusal contact; 2: with occlusal contact).

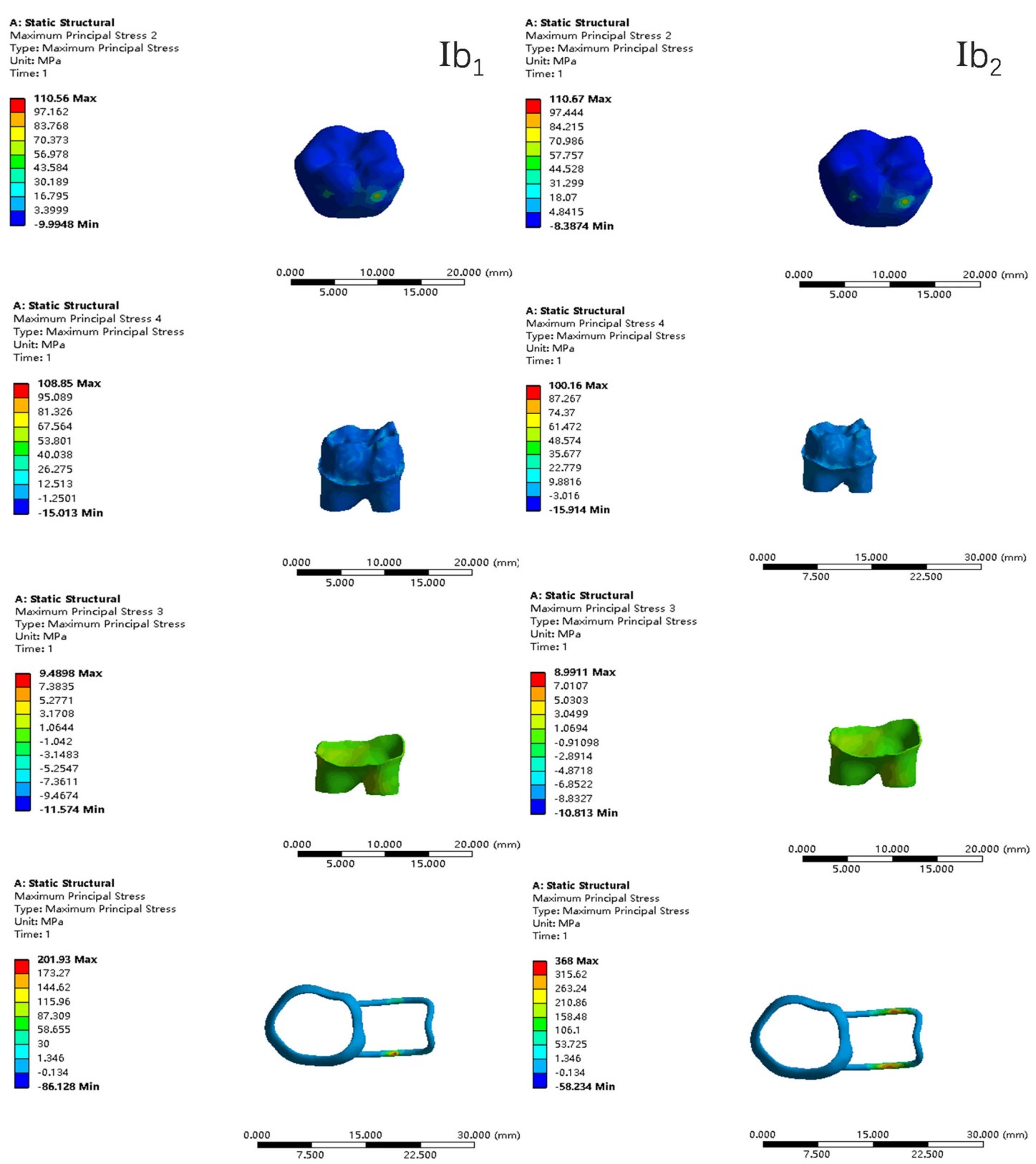

**Figure 9** The maximum principal stress distribution clouds (I: root development of 1/2; b: oblique loading; 1: without occlusal contact; 2: with occlusal contact).

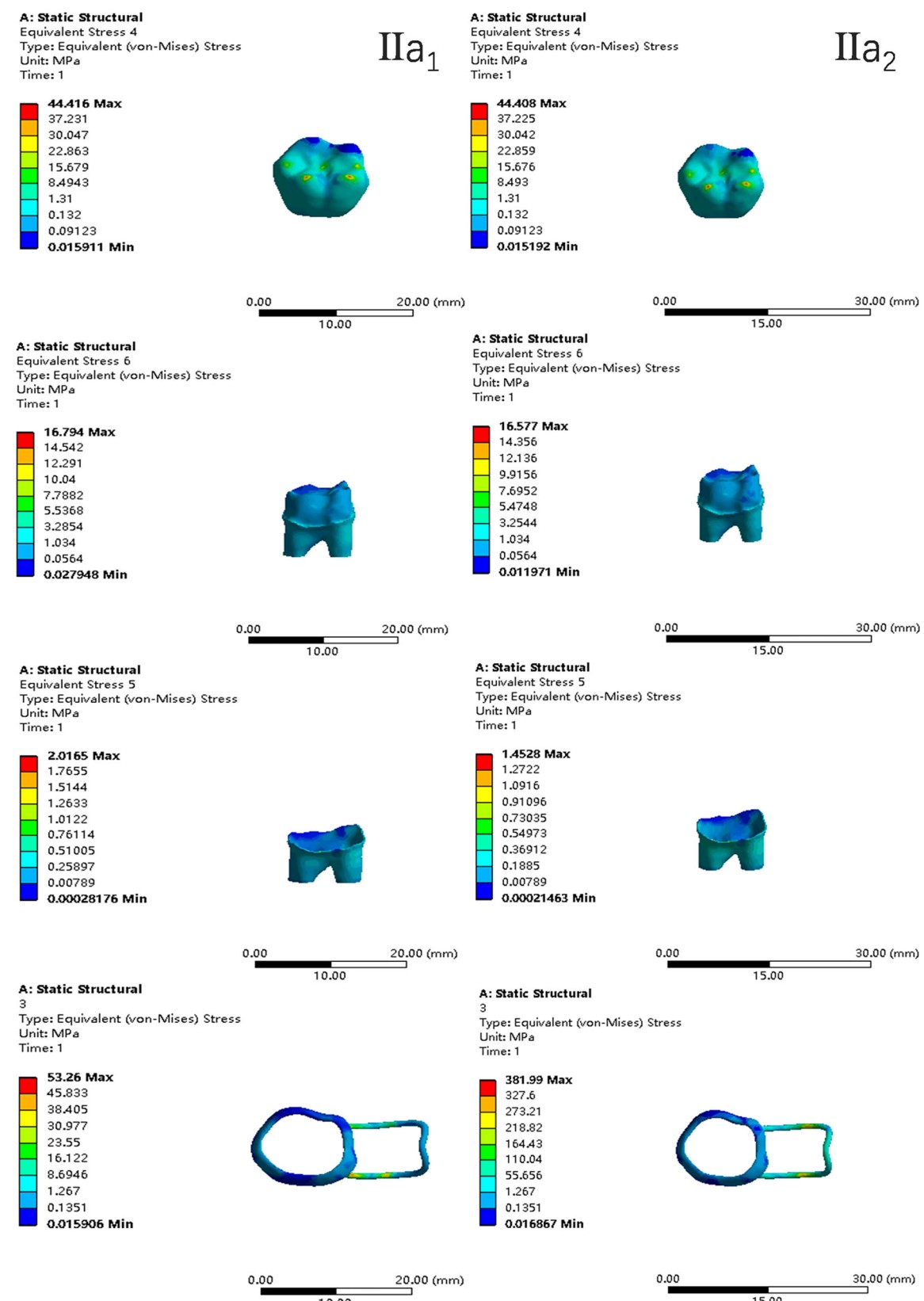

**Figure 10** The equivalent stress distribution clouds (II: root development of 2/3; a: vertical loading; 1: without occlusal contact; 2: with occlusal contact).

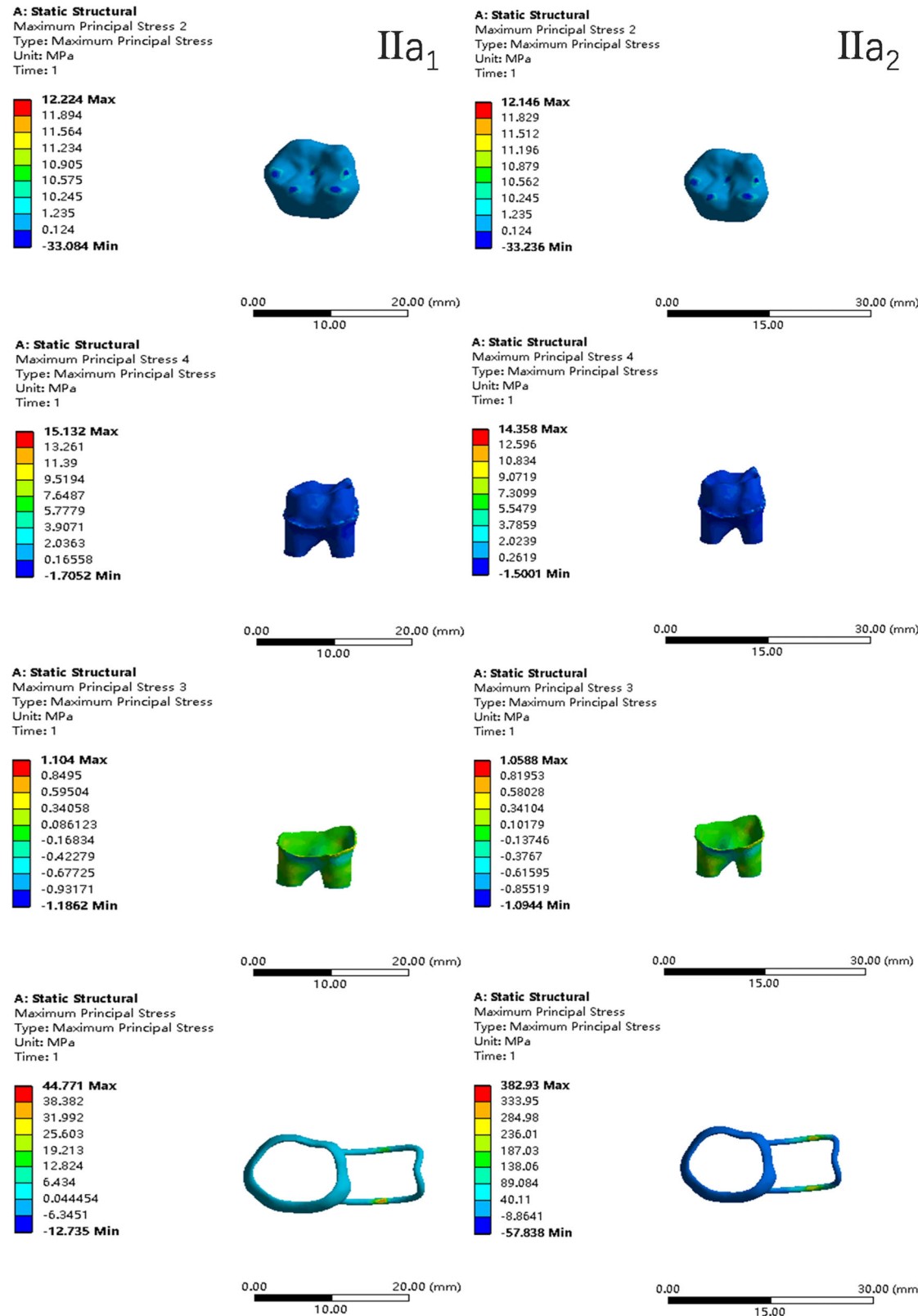

**Figure 11  The maximum principal stress distribution clouds (II: root development of 2/3; a: vertical loading; 1: without occlusal contact; 2: with occlusal contact).**

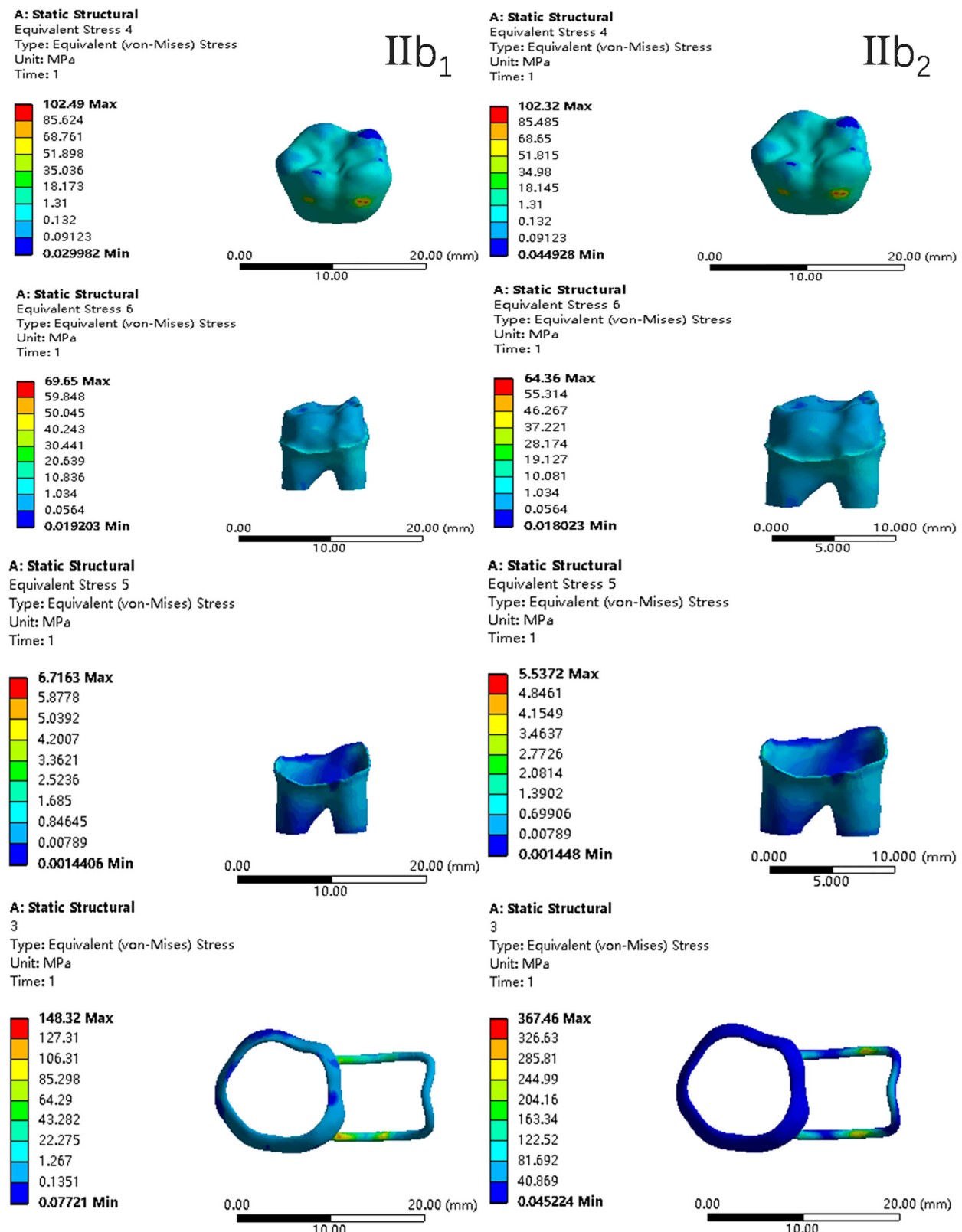

**Figure 12** **The equivalent stress distribution clouds (II: root development of 2/3; b: oblique loading; 1: without occlusal contact; 2: with occlusal contact).**

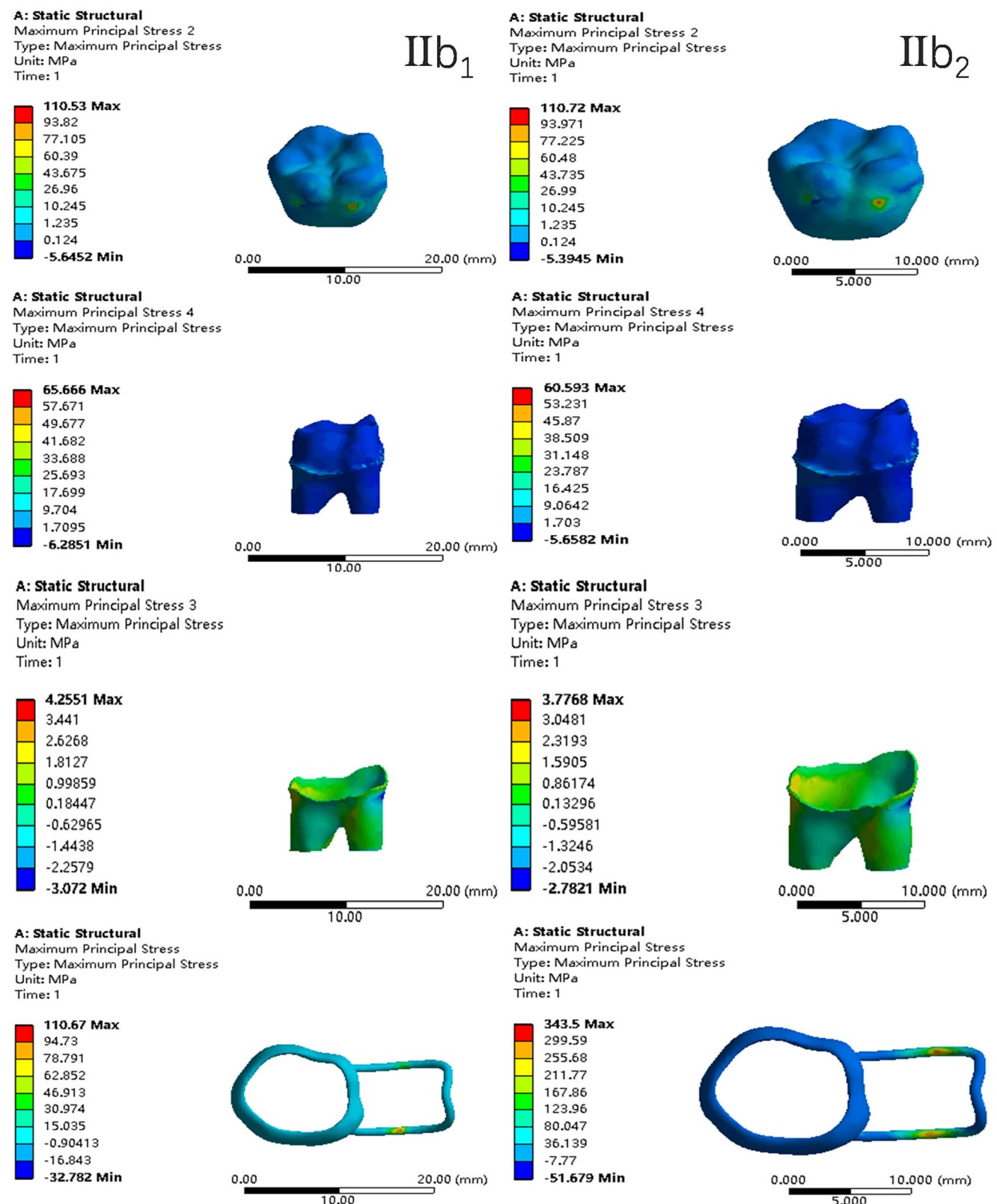

**Figure 13** The maximum principal stress distribution clouds (II: root development of 2/3; b: oblique loading; 1: without occlusal contact; 2: with occlusal contact).

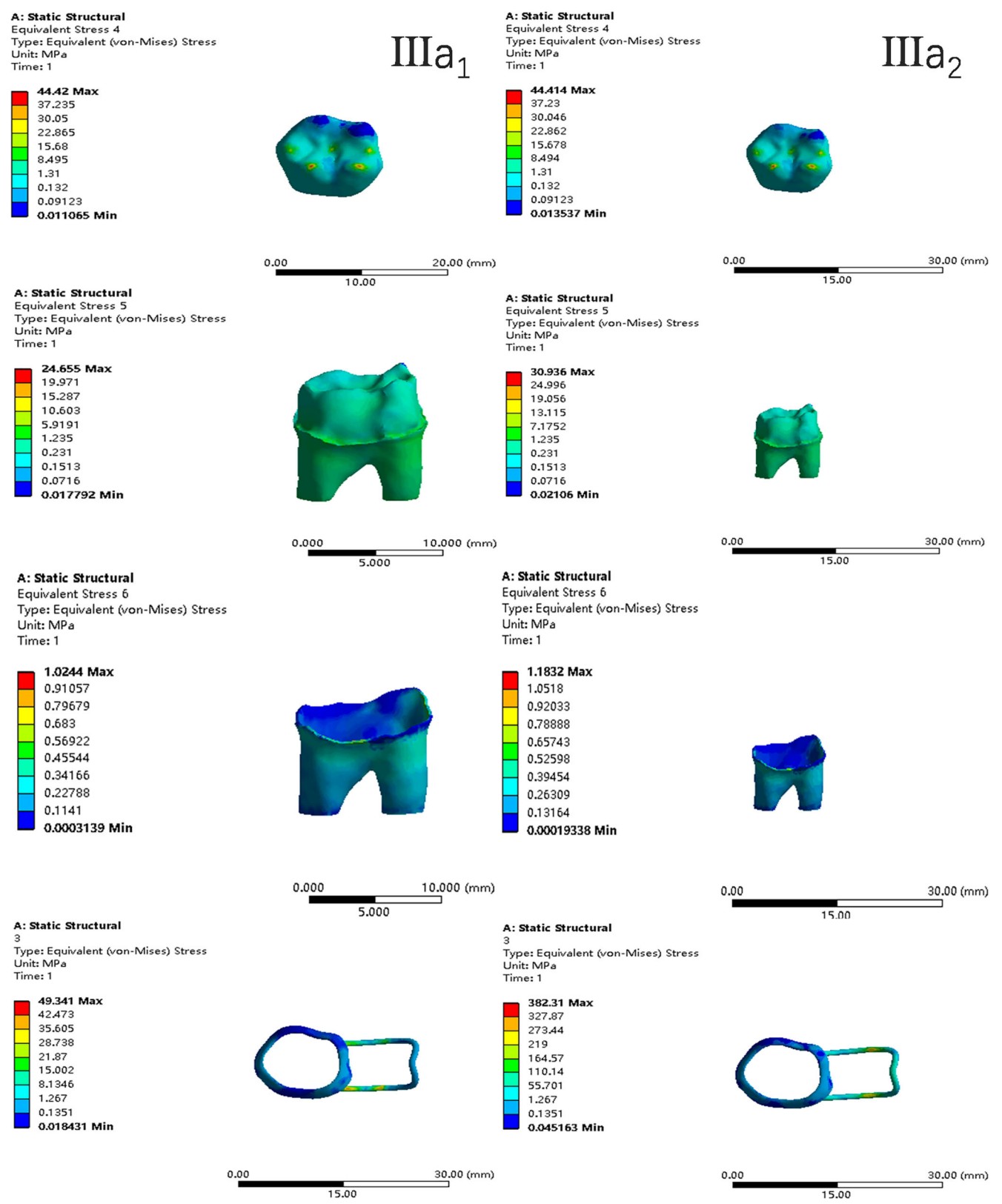

**Figure 14 The equivalent stress distribution clouds (III: root development of 3/4; a: vertical loading; 1: without occlusal contact; 2: with occlusal contact).**

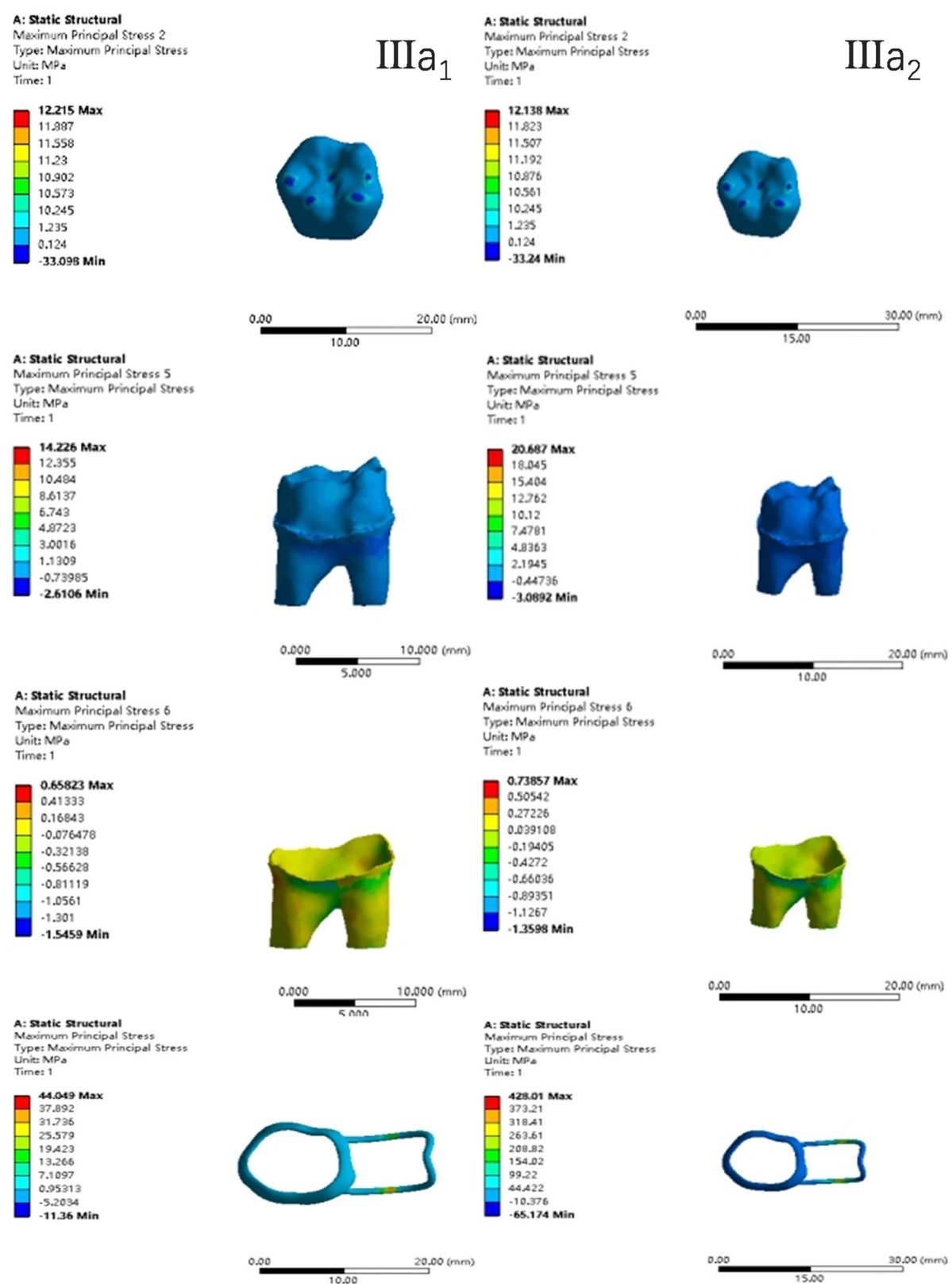

**Figure 15 The maximum principal stress distribution clouds (III: root development of 3/4; a: vertical loading; 1: without occlusal contact; 2: with occlusal contact).**

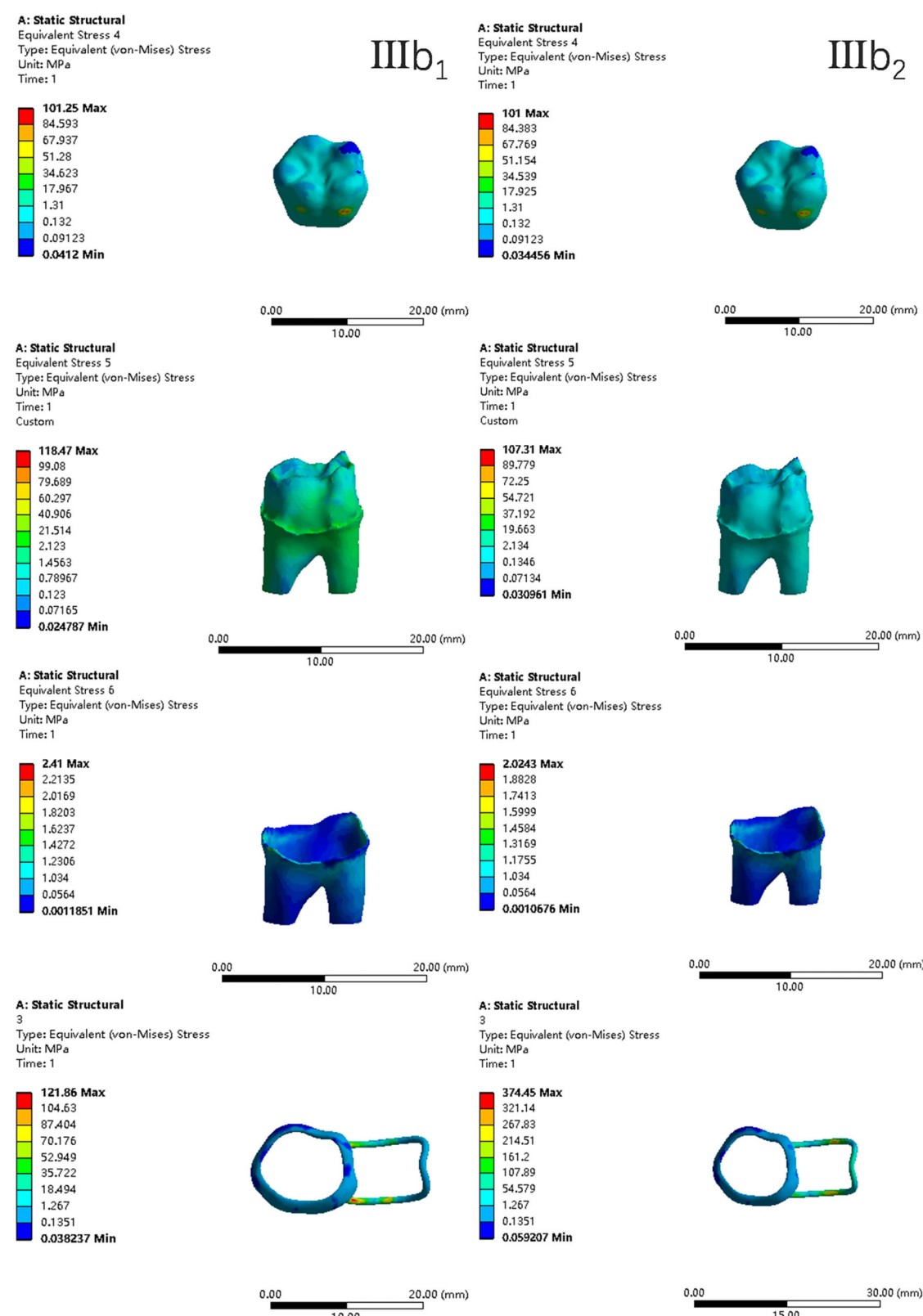

**Figure 16 The equivalent stress distribution clouds (III: root development of 3/4; b: oblique loading; 1: without occlusal contact; 2: with occlusal contact).**

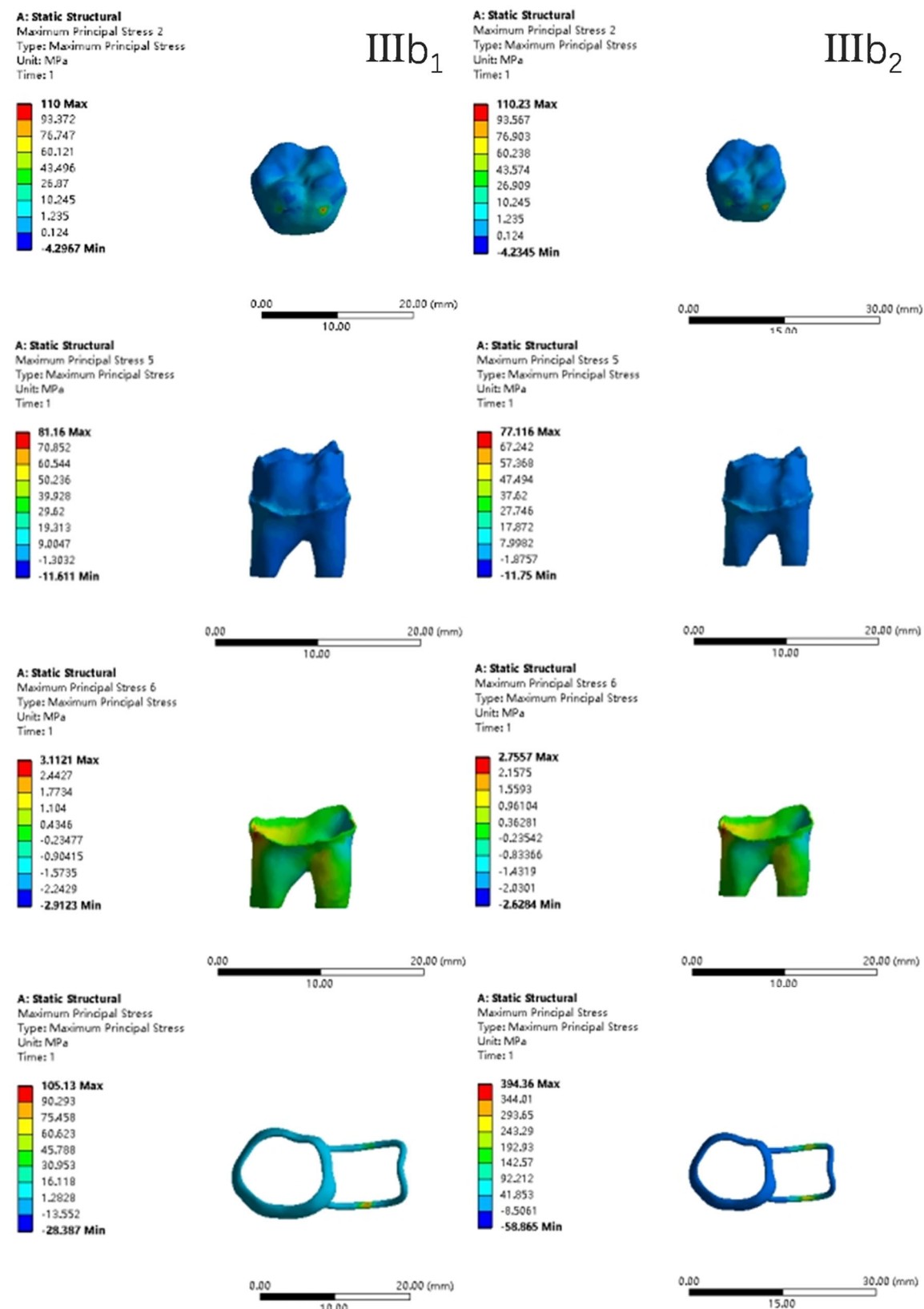

**Figure 17** The maximum principal stress distribution clouds (III: root development of 3/4; b: oblique loading; 1: without occlusal contact; 2: with occlusal contact).

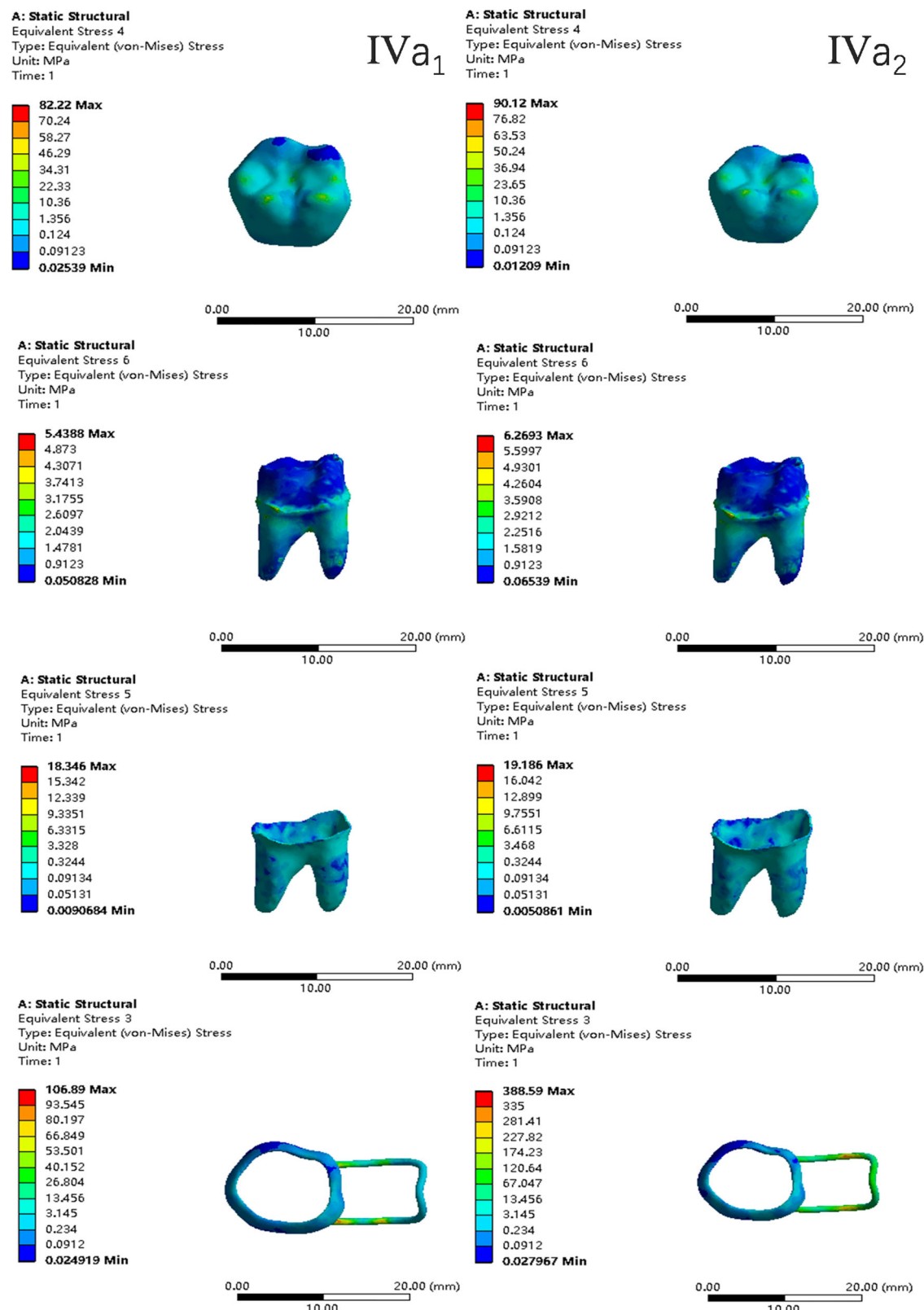

**Figure 18 The equivalent stress distribution clouds (IV: fully developed; a: vertical loading; 1: without occlusal contact; 2: with occlusal contact).**

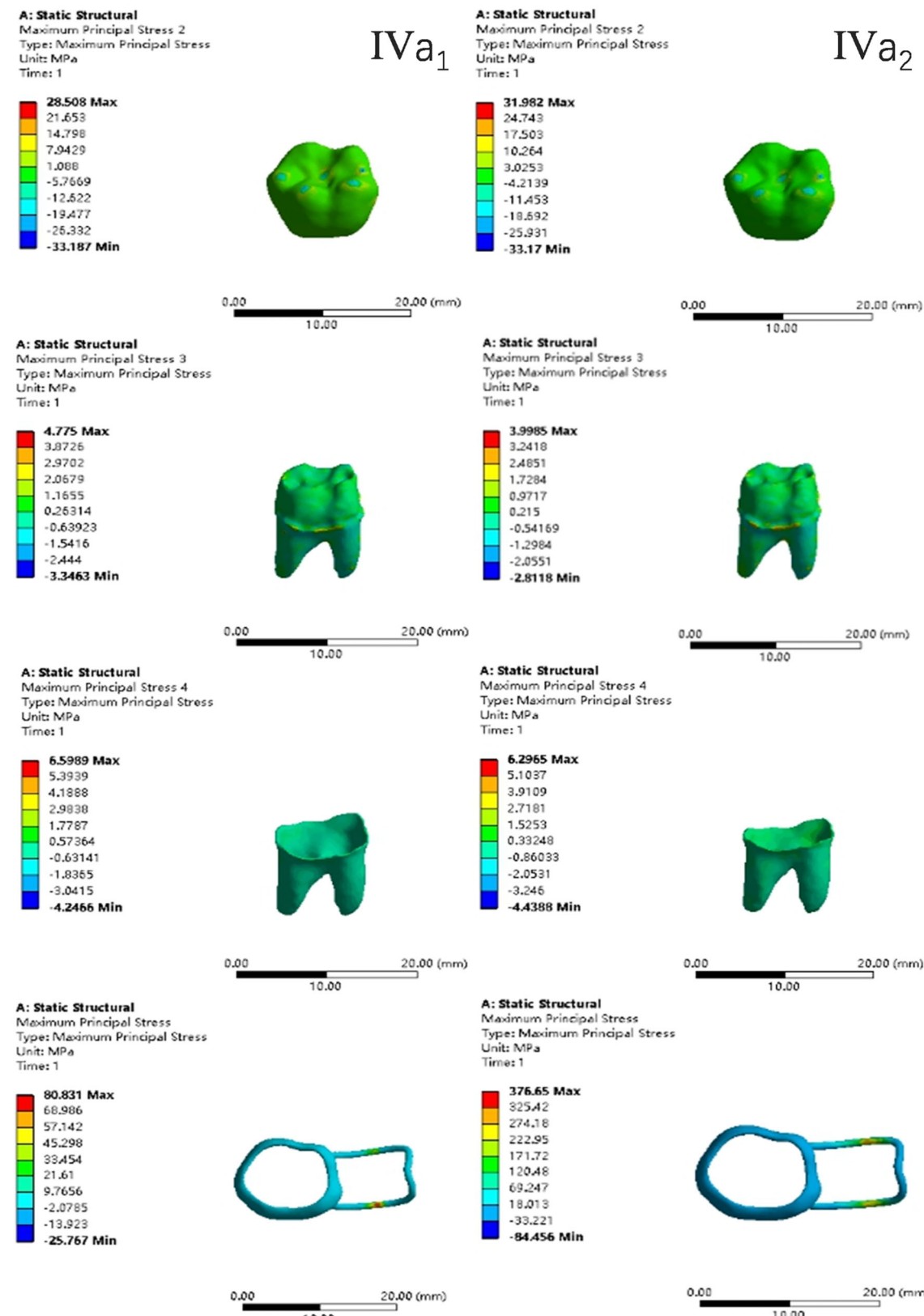

**Figure 19 The maximum principal stress distribution clouds (IV: fully developed; a: vertical loading; 1: without occlusal contact; 2: with occlusal contact).**

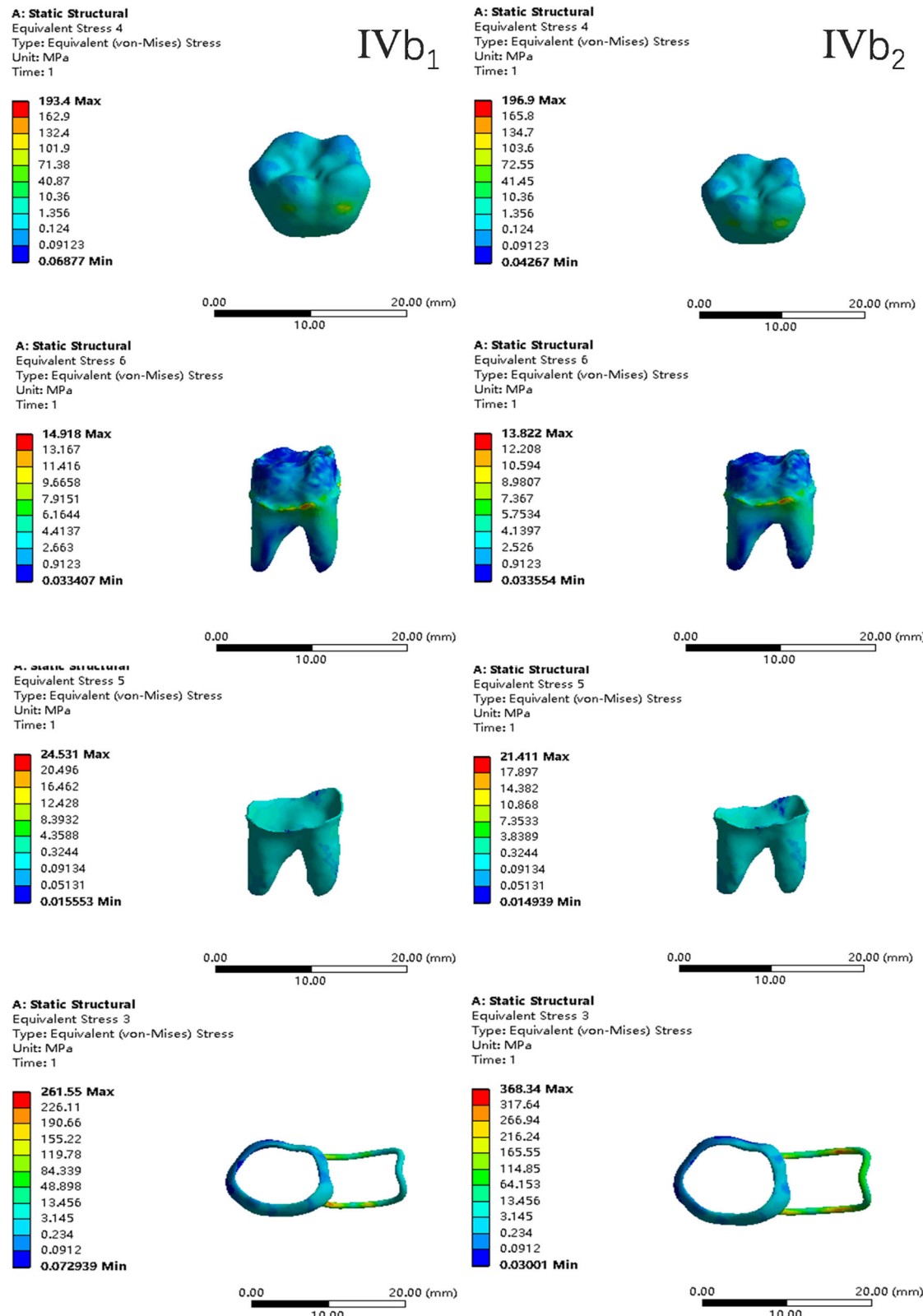

**Figure 20 The equivalent stress distribution clouds (IV: fully developed; b: oblique loading; 1: without occlusal contact; 2: with occlusal contact).**

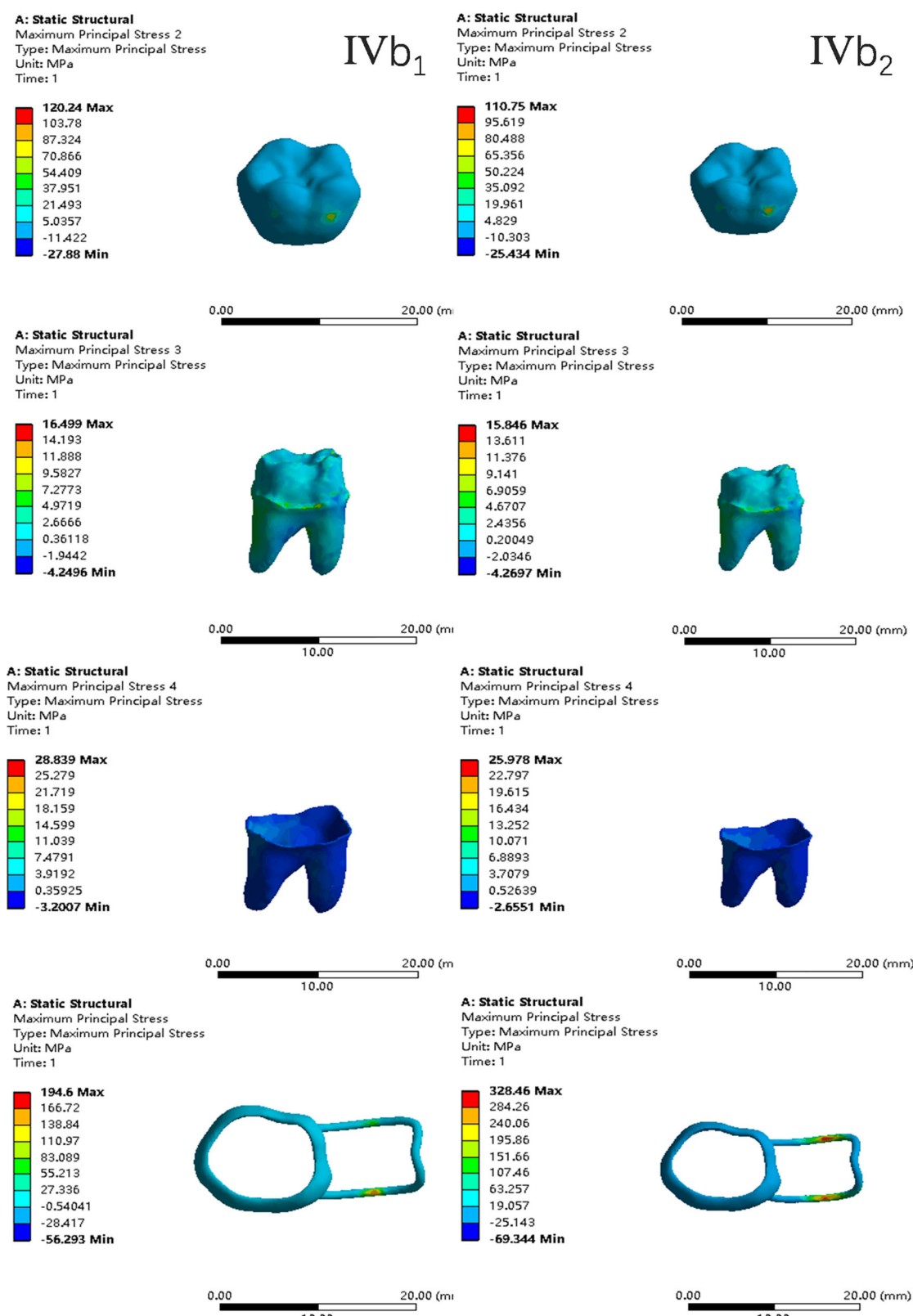

**Figure 21 The maximum principal stress distribution clouds (IV: fully developed; b: oblique loading; 1: without occlusal contact; 2: with occlusal contact).**

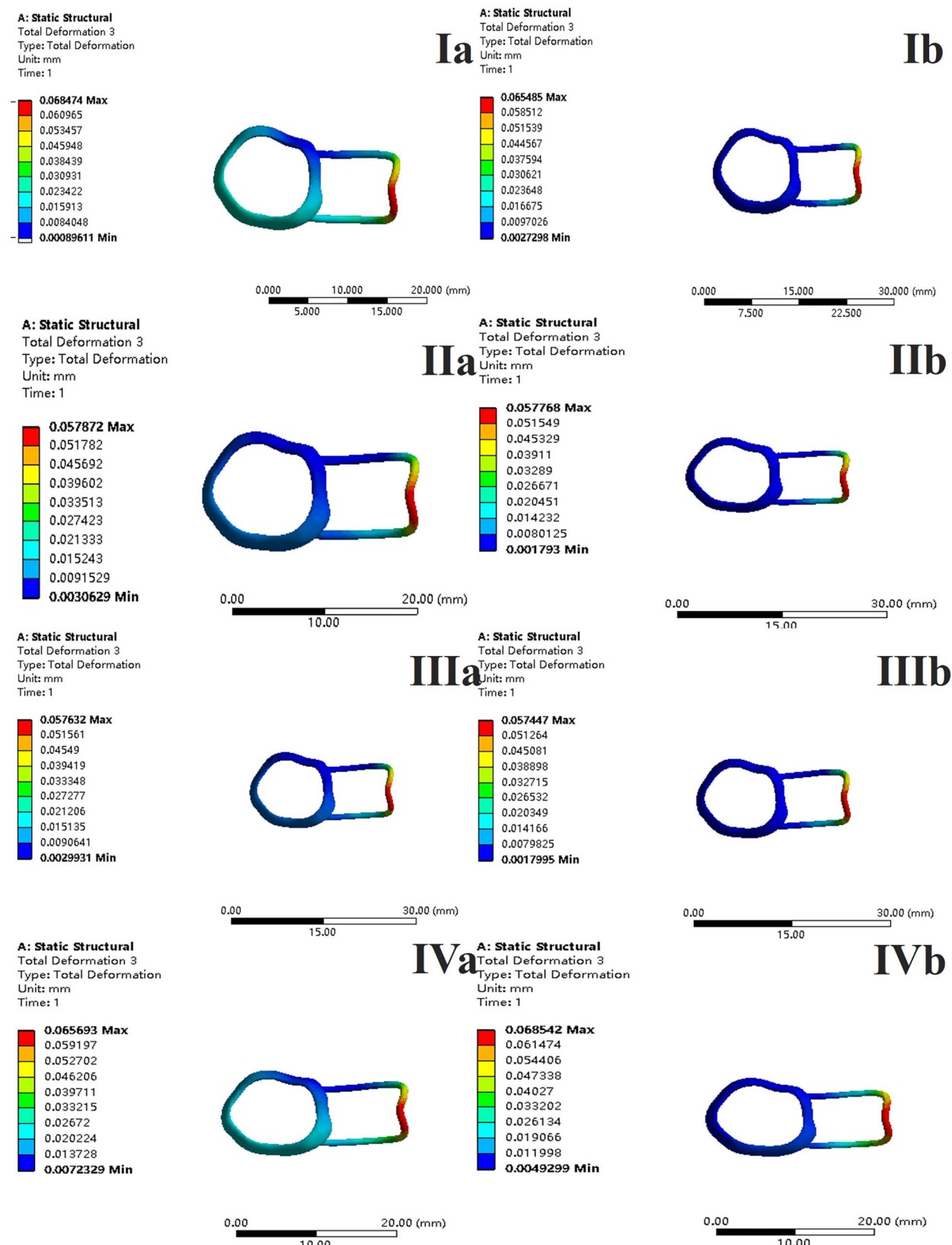

**Figure 22 With occlusal contact, displacement vector plot of band and loop space maintainer.** (A) Oblique loading; (b) vertical loading.

with the occlusal teeth. Similar research has not been conducted. The finite element analysis method is suitable for the simulation modeling and stress analysis of the jawbone, teeth, periodontal tissue, and oral materials, and can objectively and accurately reflect the stress distribution in the oral cavity by providing numerical results, offering a reference for solving clinical problems.

Based on the CT data of healthy 7-year-old children, a 3D model of healthy and standard young first permanent molars of the jaw was established, as well as a 3D model of incomplete root development. The width of the periodontal membrane is generally 0.15–0.38 mm, is narrowest near the central fulcrum of the tooth root, and wider near the alveolar crest and root apical hole, which is difficult to replicate in the model. For the purpose of this experiment, the periodontal membrane structure in the model is relatively simple, and has a uniform width of 0.2 mm wrapped around the tooth root. The research model reflects the fine structure of tooth tissue, with clear hierarchies, fine unit division, and good geometric similarity in spatial structure and anatomical morphology, which is similar to and representative of the real tooth structure.

The bite force and maximum bite force of 6–12-year-old normal children during simulated mastication were 50–100 N and 140–393 N, respectively, and the maximum axial load when chewing and swallowing various foods was 70–150 N (*Chauhan, Sharma & Jain, 2016*; *Proffit & Fields, 1983*; *Fields et al., 1986*; *Tetsuya, Toshiyoshi & Hiroyuki, 2005*; *Sonnesen & Bakke, 2005*; *Castelo et al., 2007*). The average bite force was 78 N in 6–8-year-old children (*Anderson, 1956*). Therefore, 70 N was selected as the research load to simulate the normal physiological load during mastication (*Braun, Hnat & Freudenthale, 1996*; *Tanaka et al., 2016*). When simulating median occlusion, the vertical force loading points are set to the proximal mid-cheek apex, the distal mid-cheek apex, the midpoint of the proximal mid-margin crest, the midpoint of the distal mid-margin crest, and the fossa. When simulating lateral occlusion, the oblique force loading point is set to the midpoint of the inclined cheek of the near and far middle buccal tips.

In a clinical setting, there are differences in the manufactured band and loop space maintainers, and abnormalities when used in children, which may lead to damage of the space maintainers (*Nikhil, Jyotika & Prerna, 2016*). When the space maintainer is improperly made and the bonding is not in place, direct occlusal contact between the mesial marginal crest of the first deciduous molar and the edge of the loop may occur. Additionally, the influence of improper use and eating habits on the loop is minimal so the force of the loop was also simulated at 14 N in this article. The occlusal contact between the loop and the jaw teeth was taken into consideration, and its influence on the teeth and the loop was analyzed.

In finite element analysis, the most important indicators analyzed are the equivalent stress and the maximum principal stress. Equivalent stress, or Von Mises stress, mainly examines the combined stress on the material in all directions. In finite element analysis, the comprehensive force conditions at a certain point within the material and the position of stress concentration in the model are best reflected by equivalent stress. Maximum Principal Stress reflects the maximum tensile stress in different directions at a point inside
the material, which is used to describe the actual force of the structure. Its magnitude determines whether the material can be cracked or damaged by sheer force.

Among the maximum principal stress peaks of the four groups of loops subjected to both occlusal and non-occlusal force, only the root development of the group experiencing vertical and oblique loading of the dentin falls below the tensile strength of dentin 40 MPa. Additionally, the maximum principal stress under oblique load is greater than the maximum principal stress under vertical load, which is similar to the force analysis results of previous studies on the dentin of mature mandibular first permanent molars. The developed first permanent molars model established in this study is effective and reliable. It can be used for further research in subsequent experiments. From the perspective of dentin stress, when selecting abutment teeth for band and loop space maintainers, fully developed roots are ideal.

*Chauhan, Sharma & Jain (2016)* speculated that the mature root is slenderer than the immature root, providing a larger surface area for stress transfer so the stress is distributed on the whole root surface. However, in developing teeth, only a small surface area of the root can be used to transfer the same load to the lower jaw, which means the stress is concentrated on the peri-root membrane. This contradicts the conclusion of this article which states that the stress on the periodontal ligament is greater when the root is fully developed, compared to when it is underdeveloped. This discrepancy stems from the finite element analysis which is the comprehensive result of the whole force and cannot be explained by conventional physics.

When the bite force is applied on the loop, the maximum displacement occurs at the edge of the contact between the loop and the first deciduous molar. It can be inferred that when there is too much bite force on the loop, the contact edge between the loop and the adjacent teeth may produce vertical displacement, leading to the deformation or sinking of the loop. In clinical practice, the deformation or sinking as well as the loosening and detachment of the loop due to deformation is frequently observed. When the loop is not in the occlusal contact area, the overall force of the band and loop space maintainer is observed, and the stress is concentrated at the edge of the contact between the band and the loop; when the loop is in the occlusal contact area, the stress is concentrated in the bend of the middle of the loop. The force of enamel, dentin, periodontal ligament, and alveolar bone in the four stages of root development were not significantly different between the two cases where there was no occlusal contact between the loop and the jaw teeth, and there was no significant difference in the forces of enamel, dentin, periodontal ligament, and alveolar bone. These results demonstrate that the band and loop space maintainer and the loop's bending design change the conduction direction of the force to a certain extent. This redistribution of force helps alleviate stress on the root and periodontal ligament by acting as a stress interrupter. It also prevents the first permanent molars from producing mesial displacement, therefore maintaining the edentulous space. This study suggests that occlusal contact between the loop and the jaw teeth should be avoided, and the stress break design should be added during the clinical manufacturing and bonding of the band.

## CONCLUSIONS

In this study, we examined the stress on the space maintainer and the first permanent molars. We focused on cases with missing mandibular second deciduous molars, utilizing the first permanent molars as abutments in situations with or without occlusal contact between the loop and the opposite jaw teeth during different stages of root development. However, there are still some limitations. This study assumes that the periodontal ligament is a continuous linear structure, but it is a nonlinear fiber structure. Considering the time factor, the long-term effect of a normal masticatory load needs to be further studied. Within the limited scope of this study, the following conclusions can be drawn:

(1) The stress interruption design should be included when making the space maintainer.
(2) During the design and bonding of the space maintainer, occlusal contact between the loop and the jaw teeth should be avoided.
(3) The closer the root development is to maturity, the more favorable it is to the later development of the dental arch.

## ACKNOWLEDGEMENTS

We wish to thank the timely help given by Ms Yin Zou in analyzing the number of the study.

### Funding

This work was supported by the Foundation (MS22022072), and the Top Talent Support Program for young and middle-aged people of Wuxi Health Committee (No. HB2023093) and the Young project of the Wuxi Health Committee (No. Q202102). The funders had no role in study design, data collection and analysis, decision to publish, or preparation of the manuscript.

### Grant Disclosures

The following grant information was disclosed by the authors:
Foundation: MS22022072.
Top Talent Support Program for young and middle-aged people of Wuxi Health Committee: HB2023093.
Young project of the Wuxi Health Committee: Q202102.

### Competing Interests

The authors declare that they have no competing interests.

### Author Contributions

• Hui Shi analyzed the data, authored or reviewed drafts of the article, and approved the final draft.

- Fang Fang Kang conceived and designed the experiments, performed the experiments, prepared figures and/or tables, authored or reviewed drafts of the article, and approved the final draft.
- Qian Liu analyzed the data, prepared figures and/or tables, authored or reviewed drafts of the article, and approved the final draft.

## Data Availability

The data is available at figshare: Kang, Fang Fang (2024). models.zip. figshare. Dataset. https://doi.org/10.6084/m9.figshare.24591537.v1.

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
