# Peer review of "Stress induced on permanent mandible first molar and space maintainer under normal masticatory forces: a finite element study"

_PeerJ, doi:10.7717/peerj.17456_

## Round 0.1 · original submission · Major Revisions

Your manuscript presents valuable insights into stress distribution in space maintainers and first permanent molars, utilizing finite element analysis. However, to enhance clarity, scholarly context, and overall quality, I recommend addressing the specific points raised by the reviewers in detail. I believe that these revisions will significantly improve the manuscript and look forward to receiving your revised submission.

Additionally, Please improve the figure quality and make more descriptions in figure legends.

Reviewer 1 ·

Basic reporting

1.References No. 3 and No. 7 are duplicates and should be revised.
2.There are few references in this article, can you cite more?

Experimental design

1.This article study the stress on the first permanent molar and the loop with or without occlusal contact, with the first permanent molar of four different degrees of development serving as the abutment.. Whether it is necessary to consider the case where the root of the tooth is less than one-half developed.
2.How does the first conclusion relate to the content of this paper?
3.The content of the article is somewhat innovative, please describe it more.
4.Please add what are the requirements for selecting the CT data required to build the model.
5.The pixels of the stress contour map need to be sharper.
6.The introduction lacks a description of the undesirable bite force withheld by the loop.

Validity of the findings

1.The research perspective of this article is relatively novel.
2.The article has certain research value and clear logic.

Reviewer 2 ·

Basic reporting

This study aims to provide valuable insights into the design and use of space maintainers for maintaining missing spaces of deciduous molars.
The authors mentioned the application of the band and loop space maintainer in maintaining the space lost early by deciduous molars, as well as the potential stress on the first permanent molar during the mixed dentition stage when the root is still developing. However, the authors could further supplement the importance of finite element analysis method and how it is utilized to study the stress distribution between teeth and space maintainers.

Experimental design

(1) please provide a workflow to illustrate the steps and processes described in the methods. Adding a flowchart to the methods section may make the entire study more readable and visually appealing.
(2) Using only words to describe the equipment in section 1.1 may not be sufficient as it lacks specificity and clarity. It is recommended to provide a more detailed description of the equipment, including the brand, model, and specifications, to ensure that readers have a clear understanding of the tools and resources used in the study.
(3) Having a complete sentence as a subheading in section 1.3 may not be ideal as subheadings are typically concise and serve as a brief summary of the content that follows. It is recommended to use short phrases or keywords.
(4) To simplify the cumbersome numbering like 1.3.2.1 and 1.3.2.2, you may consider using more concise labeling, such as using alphabetical markers (e.g., 1.3.2.a and 1.3.2.b) or directly using subheadings (e.g., Part 1 and Part 2 within 1.3.2).
(5) Please rename the subheading “methods”.

Validity of the findings

Even with comprehensive tables and graphical support, authors should describe key results in the text, providing sufficient detail to facilitate a comprehensive understanding of the research findings. This approach adheres to best practices in scientific writing, ensuring transparency and replicability of the research.
Alongside presenting numerical data, the Results section should offer a brief interpretation of what these numbers mean in the context of the study's objectives. For instance, if the study finds that the stress on the space maintainer is significantly lower at a certain stage of root development, this finding should be clearly stated and its potential implications briefly discussed.

Additional comments

Please compare and contrast the study's results with those from previous research, highlighting similarities and differences.
The figures are not very clear, please improve the quality of the graphics and provide detailed descriptions in the figure legends.

---

## Round 0.2 · accepted · Accept

All comments were addressed by authors, and this manuscript has been significantly improved. This paper has been more readable after revisions and I think it is suitable for publication.

Reviewer 1 ·

Basic reporting

References have been corrected by authors. This article is well written and many pictures are presented.

Experimental design

All my concerns were addressed by authors.

Validity of the findings

This is an interesting study with a lot of work. No any other questions.

Reviewer 2 ·

Basic reporting

no comment

Experimental design

no comment

Validity of the findings

no comment

Additional comments

The revised manuscript has been significantly improved. The expanded description of the experiments enhances the clarity and reproducibility of the research. The upgraded quality of the figures and tables greatly improves the visual presentation and readability of the paper. The inclusion of source file links demonstrates research transparency and allows readers to further explore the data and methodological details.
These improvements have substantially addressed the concerns raised in the previous review and elevated the manuscript to a publishable standard. Therefore, I am pleased to recommend the acceptance of this paper.